# “Stop, Little Pot” as the Motto of Suppressive Management of Various Microbial Consortia

**DOI:** 10.3390/microorganisms12081650

**Published:** 2024-08-12

**Authors:** Elena Efremenko, Nikolay Stepanov, Olga Senko, Olga Maslova, Ilya Lyagin, Maksim Domnin, Aysel Aslanli

**Affiliations:** Faculty of Chemistry, Lomonosov Moscow State University, Lenin Hills 1/3, Moscow 119991, Russiasenkoov@gmail.com (O.S.);

**Keywords:** metal nanoparticles, mycoviruses, antimicrobial peptides, quorum molecules, polymicrobial biofilms, filamentous fungi, yeasts, combinations of antimicrobials, enzymes, adhesion

## Abstract

The unresolved challenges in the development of highly efficient, stable and controlled synthetic microbial consortia, as well as the use of natural consortia, are very attractive for science and technology. However, the consortia management should be done with the knowledge of how not only to accelerate but also stop the action of such “little pots”. Moreover, there are a lot of microbial consortia, the activity of which should be suppressively controlled. The processes, catalyzed by various microorganisms being in complex consortia which should be slowed down or completely cancelled, are typical for the environment (biocorrosion, landfill gas accumulation, biodegradation of building materials, water sources deterioration etc.), industry (food and biotechnological production), medical practice (vaginitis, cystitis, intestinal dysbiosis, etc.). The search for ways to suppress the functioning of heterogeneous consortia in each of these areas is relevant. The purpose of this review is to summarize the general trends in these studies regarding the targets and new means of influence used. The analysis of the features of the applied approaches to solving the main problem confirms the possibility of obtaining a combined effect, as well as selective influence on individual components of the consortia. Of particular interest is the role of viruses in suppressing the functioning of microbial consortia of different compositions.

## 1. Introduction

Today, the resistance of microorganisms to antimicrobial agents used in medicine and veterinary medicine [1], various industries [2], agriculture [3] and other areas [4] is widely discussed throughout the world. To solve this problem, researchers are searching for new antimicrobials that act on individual types of cells, but the difficulty of achieving this goal in practice lies in the fact that in nature, cells most often exist as part of polymicrobial consortia. Their microbial composition allows consortia to realize the total metabolic potential of cells and overcome the effects of antimicrobial agents, as well as other suppressive methods of influence (increased temperature, pressure, changes in pH, etc.), including through the formation of protective heterogeneous biofilms [5,6]. The composition of such biofilms may include pathogenic bacteria, yeasts, filamentous fungi and microalgae [5], which are often powerful destructors of various xenobiotics, including antimicrobial agents [6,7]. Among these pathogens that can cause various systemic diseases, *Candida*, *Aspergillus*, *Fusarium*, *Pseudomonas*, *Staphylococcus*, and *Klebsiella* are especially dangerous [1,2,7]. Moreover, high concentrations of cells in such biocommunities lead to an increase in the pathogenicity of these cells due to the manifestation of the results of quorum sensing (QS) and emergence [7]. In addition, it is known that quorum molecules synthesized by some cells, for example, bacteria, can have an overlap and activating effect on other microorganisms, for example, microalgae that synthesize microalgae toxins, helping to increase the resistance of the formed consortia to the antibiotics exposure and increasing their danger [8,9].

Microorganisms within biofilms can coexist in separate microcolonies with limited interactions while being mixed or having a layer-by-layer arrangement when one type of cell is located in the upper layer and another type in the lower [10,11] (Figure 1A–C). At the same time, different microconditions are created in biofilms in terms of oxygen saturation (aerobic, microaerophilic, anaerobic), as a result of which biosystems that are different in characteristics and difficult to be effectively influenced are formed. Additionally, horizontal and vertical gene transfer between different types of microorganisms, which contributes to the emergence of bacteria and fungi with new properties that can increase their virulence and resistance to antimicrobial drugs [12]. In biofilms, yeast cells undergo a transition to filamentous form with an increase in the synthesis of proteases, which is necessary for their better adhesion to various surfaces [13]. In polymicrobial consortia, individual bacteria play a key role in regulating the transition of yeast to filamentous form. The interaction of *Candida albicans* with other microorganisms can occur through co-aggregation and co-adhesion. For example, the interaction of cells of the genus *Candida* with *Streptococcus* cells is mediated by polysaccharide receptors on the surface of bacterial cells and adhesins of *C. albicans* [14]. In addition, *Streptococcus* cells produce lactate, which is consumed by yeast, while yeast quorum molecule (farnesol) in mixed biofilms enhances the growth of *Streptococcus mutans* cells.

Biofilms, consisting of cells of the bacteria *Staphylococcus aureus* and the yeast *C. albicans*, are often observed in diseases of the mucous membranes, including the oral cavity. There are mixed biofilms of different *Candida* species, for example, *C. albicans* and *C. glabrata*, *C. krusei*, together with several species of bacteria, including *Staphylococcus*, *Streptococcus*, *Klebsiella* and *Pseudomonas*. For example, the *Streptococcus gordonii* bacteria are able to enhance the development of hyphae in *C. albicans*, leading to the formation of a polymicrobial biofilm [15]. In addition to yeast-bacterial infections, the number of invasive lesions of human and animal tissues by pathogenic filamentous fungi is growing. For example, Aspergillus and Fusarium are plant pathogens that cause food spoilage and synthesize mycotoxins [16]. Using the example of the cells of the fungus *Fusarium graminearum* it was found that the process of biofilm formation begins with the attachment of spores to various surfaces, followed by cell proliferation and the formation of a matrix consisting of polysaccharides and proteins, followed by abundant formation and spread of conidia [17] (Figure 1D,E). In addition, filamentous fungi secrete special proteins (hydrophobins), which are involved in the adhesion of hyphae to hydrophobic surfaces and can contribute to the formation of biofilms. Bacteria promote the transition of fungal cells from vegetative growth to the induced formation of spores (conidia), which are included in biofilms [18] (Figure 1D). Interactions between filamentous fungi and bacteria occur through their attachment and growth on hyphae and can lead to changes in the production of secondary metabolites. For example, *Bacillus subtilis* in the presence of *Aspergillus* cells reduced the synthesis of surfactin, and the presence of *Streptomyces* cells in a joint consortium with *A. flavus* led to a decrease in the accumulation of aflatoxin B [19].

Fungi make an important contribution to the functioning of activated sludges used for wastewater treatment under aerobic and anaerobic conditions [20]. Fungi of the genera *Pluteus*, *Wickerhamiella* and *Penicillium* are the most widespread in activated sludge. It has been proven that dissolved oxygen and the C/N ratio are the most significant factors influencing changes in the structure of the fungal community of activated sludge ([21], Figure 1F).

Consortia, with the participation of fungi, are known to be biocatalysts of negative processes associated with the development of biocorrosion [22] or the release of landfill gases [23]. Dangerous communities of fungi (mycelial and yeast-like), bacteria and microalgae include consortia that have a destructive effect on buildings and monuments [24,25,26,27]. At the same time, the filamentous fungi are active producers of various hydrolytic enzymes, which provide a significant part of the microorganisms of the consortia with available nutrients from the organic sources present, thus triggering the overall metabolism of microbial communities. This is especially significant for landfills and methanogenic consortia [23].

Thus, given the high level of resistance of polymicrobial consortia to used antimicrobial drugs, the search for approaches and drugs to suppress the development of such complex biosystems involved in various processes is extremely relevant. At the same time, knowledge about methods of suppressing individual participants in such consortia containing specifically fungal cells and identifying objects that are most vulnerable to suppressive effects are necessary to ensure the effectiveness of suppressive actions (Figure 2).

At the same time, the knowledge gained can be extrapolated to other consortia, the activity of which must be eliminated in those processes where they cause a negative effect. This review is aimed at analyzing approaches to the suppression of polymicrobial consortia and their effectiveness. Publications carried out over the past 10 years that focused on the suppression of polymicrobial consortia were taken into account mainly to analyze existing developments and trends in their development. The articles were searched in several scientific databases (https://scholar.google.com, https://pubmed.ncbi.nlm.nih.gov—the last accessed 1 June 2024) by using combinations of the following keywords: mixed polymicrobial consortia, mixed biofilms, fungal consortia, etc.

## 2. Various Compounds and Their Combinations against Polymicrobial Consortia

To date, many antimicrobial drugs and methods have been developed in consortia to combat biofilms and pathogens over the past 10 years ([23,24,25,26,27], Table 1 [28,29,30,31,32,33,34,35,36,37,38,39,40,41,42], Table 2 [43,44,45,46,47,48,49], Table 3 [50,51,52,53,54,55,56,57,58,59,60,61,62,63,64,65,66,67], Table 4 [68,69,70,71,72,73,74,75,76,77,78], Table 5 [79,80,81,82,83,84], Table 6 [85,86,87,88,89,90,91,92], Table 7 [93,94,95,96,97,98,99,100,101,102,103,104]).

The main attention is paid to various approaches based on the destruction of the biofilm matrix and increasing its permeability, preventing the adhesion of biofilms to different surfaces, and also focused on cell death in consortia [105]. For this purpose, biologically active compounds of plant and animal origin [28,29,30,31,32,33,34,35,36,37,38,39,40,41,42], antimicrobial peptides [43,44,45,46,47,48,49], various antimicrobial drugs and disinfectants [50,51,52,53,54,55,56,57,58,59,60,61,62,63,64,65,66,67], heavy metal ions and metal NPs [25,26,68,69,70,71,72,73,74,75,76,77,78], physicochemical effects [79,80,81,82,83,84], enzymes [23,85,86,87,88,89,90,91,92], microorganisms and their metabolites [27,93,94,95,96,97,98,99,100,101,102,103,104], as well as combinations of various compounds and methods [23,26,41,42,47,48,49,65,66,67,74,75,76,84,89,102,103,104] are used (Figure 2).

This approach is focused on reducing the doses of used substances with antimicrobial activity and reducing the toxicity they bring. When implementing a combined approach to suppressing the growth and metabolic activity of microbial cells in mixed consortia, the main questions arise about what is best to combine, what does not have a significant effect on polymicrobial biosystems, and how the composition of the latter affects the success of the suppressive effects. The subsequent sections of this article are devoted to finding answers to these questions.

### 2.1. Effects of Non-Peptide Antimicrobial Compounds of Plant and Animal Origin on Mixed Consortia

Today, a number of plant substances of non-peptide nature are known (Table 1 [28,29,30,31,32,33,34,35,36,37,38,39,40,41,42], Figure 3), such as myrtenol [30], allicin [31,32], nepodin [33], etc., which inhibit the expression of a number of genes responsible for various processes in cells included in yeast-bacterial consortia. For example, there is inhibition of
-haemolysin genes (hla and hld) in *S. aureus*, several biofilm-related genes in *E. coli* (csgAB, fimH and flhD) and hypha cell wall gene HWP1 in *C. albicans* [28];-genes responsible for the synthesis of substances involved in the formation of QS in *P. aeruginosa* [29];-genes responsible for the motility of *Klebsiella pneumoniae* bacteria, their adhesion and biofilm formation (genes mrkA, FKS1, ERG11 and ALS5), as well as the gene ERG11 participating in the ergosterine synthesis and gene FKS1 dealing with β-1,3-glucane synthase which is a key enzyme participating in the production of main polysaccharide component of cell walls [30];-genes responsible for the formation of hyphae and biofilms (genes ECE1, HGT10, HWP1 and UME6) and regulation of transport functions (genes CDR4, CDR11 and TPO2) [33].
microorganisms-12-01650-t001_Table 1Table 1Inhibition of polymicrobial consortia by non-peptide antimicrobial compounds of plant and animal origin.ConsortiaForm and Site (Reason) of PresenceSuppressive CompoundsEffects*C. albicans*/*S. aureus*, *E. coli* [28]Biofilm host and environmental surfacesSaw palmetto oil (100 µg/mL), lauric acid and myristic acid (20 µg/mL) with dimethyl sulfoxide (DMSO)90% inhibition of bacteria/yeast biofilm formation without affecting planktonic cell growth*Pseudomonas aeruginosa*/*Aspergillus fumigatus* or *Scedosporium apiospermum* [29]Biofilms, patients with chronic infectionsPompia and grapefruit essential oils (10 mg/L) with DMSO70% inhibition of biofilm formation*Candida auris*/*Klebsiella pneumoniae* [30]Biofilms in the urinary tract, bronchi, liverMyrtenol (50 μg/mL) with DMSO90% inhibition of biofilm formation*Candida albican*/*E. coli* [31]Biofilm, mucosal surfacesAqueous garlic extract (50 mg/mL)70.2% decrease in cell concentration in biofilm *C. albicans*/*Klebsiella pneumoniae* [32]Biofilm, urinary tract, device-related infections*Allium ursinum* and *Allium oschaninii* methanol extracts (75 μg/mL)Up to 99% death of microorganisms in biofilms*C. albicans*/*S. aureus* or *Acinetobacter baumannii* [33]Biofilm, silicon catheterNepodin (10–20 µg/mL) 75–85% inhibition of biofilm formation *C.tropicalis*/*S. aureus* [34]Biofilm, nosocomial infections*Glycyrrhiza glabra* extract (1.5 mg/mL) and Manuka honey (37.5%) Decrease of cell amount inside biofilm to 1.0–3.5 log CFU/mL*C. parapsilosis*/*Exophiala dermatitidis* [35]Biofilm, infection of toenailEthanol extract of propolis (1675 μg/mL)Total eradication of *E. dermatitidis* in biofilms and 14% reduction of *C. parapsilosis* cells*C. albicans*/*E. faecalis* [36]Biofilm, tongue mucosal infections, sputum, sepsis, and root canal infectionsLuteolin (256 μg/mL) with DMSO78% death of microbial cells in biofilms and destruction of biofilm matrix components (mainly polysaccharides and proteins)*C. albicans*/*E. coli* [37]Biofilm, urinary catheters and endotracheal tubesLawsone (100 µg/mL) with DMSOReduced curli production in *E. coli* and *C. albican* hyphal formation*A. niger*, *Aureobasidium pullulans*, *Chaetomium globosu*, *Cladosporium cladosporioides Alternaria alternata*, and *Penicillium citrinum* [38]Microbial colonization, biofilm, modern painting; cultural heritage *Citrus aurantium* hydrolate (99.97% *v*/*v)* and *Cinnamomum zeylanicum* essential oil (0.03 *v*/*v*),5 h (28 μL/cm^2^)Complete killing of the fungal species; inhibition of fungal ATPases and cell wall biosynthesis by altering the membrane structure and integrity*Aspergillus* sp., *Penicillium* sp. and *Mucor* sp. [39]Microbial colonization,cultural heritage*Ocimum basilicum* hydro-alcoholic and water extracts and essential oils (15 µL/paper discs)Inhibition of fungal growth (100%) up to 144 h incubation*Hormoconis*, *Aspergillus*, *Fusarium*, *Trichosporon* [40]Biofilms, oil industryEssential oil of *Lippia gracilis* Schauer (20 μg/L)Completely inhibition of the fungal growth**Combinations of antimicrobial non-peptide compounds of plant origin with antibacterials or antifungals***C. albicans*/*S. mutans* [41]Biofilm, dental plaqueEugenol (12.5 μg/mL with azithromycin (128 μg/mL) with DMSOEradication of mixed biofilms;Fractional Inhibitory Concentration Index (FICI-0.14) *C. albicans*/*S. aureus* [42]Biofilm, skin wounds, denture stomatitis and bloodstream infectionsBerberine (128 μg/mL) with amphotericin B (4 μg/mL) with DMSO96% death of *C. albicans*,76% death of *S. aureus* cells


It has been established that flavonoids are effective inhibitors of biofilm development [31,32,34,35]. Thus, flavonoids present in garlic extracts can effectively suppress cell growth in yeast-bacterial biofilms (*C. albicans*/*E. coli* [31], *C. albicans*/*Klebsiella pneumoniae* [32]), suppressing the process of their formation by 70–90%. It was noted that it is possible to achieve the death of 99% of cells when using methanol instead of water for the preparation of a similar extract to suppress the formation of the same biofilm by *C. albicans*/*K. pneumoniae* cells [32] on teflon surfaces.

The combination of flavonoids (≈75% of the total phenolic compounds) present in the extract of *Glycyrrhiza glabra* with biologically active substances contained in bee honey showed a pronounced additive antimicrobial effect against mixed biofilms of *C. tropicalis*/*S. aureus* [34]. In the case of a consortium in which all participants were represented by yeast that provokes skin diseases, an extract of another bee product (propolis) containing flavonoids and essential oils demonstrated selective suppressive effect only against *Exophiala dermatitidis* cells (up to their complete death), without inhibiting the second member of the consortium (*Candida parapsilosis*) [35].

Lawsone (2-hydroxy-1,4-naphthoquinone), extracted from henna leaves or water hyacinth flowers, is used as a dye. However, it turned out that at a concentration of 35 µg/mL lawsone exhibits antimicrobial activity against yeast-bacterial biofilms of *C. albicans*/*E. coli*, while it also acted quite selectively, effectively inhibiting the aggregation of *C. albicans* yeast cells and the formation of hyphae by them, thereby disrupting biofilm formation [37]. It is known that secondary plant metabolites in the form of terpenoids, alkaloids and various organic acids can exhibit fungicidal properties. For example, the production of mycotoxins that protect fungi, including those in fungal consortia, is inhibited by L-pyroglutamic acid at the level of transcription of proteins necessary for their synthesis [106].

Flavonoids and essential oils containing substituted phenols, such as eugenol, thymol and carvacrol, have strong antimicrobial and antioxidant activity, including against microscopic fungi such as *Penicillium digitatum*, *P. italicum*, *P. expansum*, *F. nivale*, *F. solani*, *Colletotrichum musae* and *Lasiodiplodia theobromae* [107,108].

Interestingly, plant essential oils inhibit the growth of fungi to a greater extent than bacteria [29], although in the case of mixed consortia, the growth of both bacterial and yeast cells is inhibited, as it was in the case with nepodin [33]. It has been established that citrus essential oils inhibit the QS of *P. aeruginosa* bacteria and cause an increase in the membrane permeability of *C. albicans* yeast cells [29]. Essential oils of plants have also shown their effectiveness in suppressing the growth of filamentous fungi as part of consortia colonizing ancient paper artifacts [39].

The results of a number of studies have shown that antimicrobial agents of plant origin, in combination with known antibiotics and fungicides, demonstrate synergistic effects [41,42]. The combination of amphotericin B with berberine demonstrated a significant impairment of hyphotic filamentation of *C. albicans* and co-adhesion between yeast and bacteria and a decrease in the expression of genes associated with QS in both cell types [42]. Drugs combined with herbal compounds can act on multiple targets simultaneously. The doses of antimicrobial agents required for use are reduced in the combinations applied against consortia [41]. It is assumed that essential oils disrupt the integrity of the cell membrane, ensuring the penetration of drugs into microbial cells within consortia, which increases the effectiveness of antimicrobial agents [41].

Thus, an effective approach to the suppression of polymicrobial consortia is the use of antimicrobial compounds of plant and animal origin in combination with fungicides and antibiotics, which provides an expanded range of targets for suppressive effects.

### 2.2. Antimicrobial Peptides (AMP) against the Polymicrobial Consortia

Today, special attention in the fight against microbial infections is directed not only to the search for new antibiotics but also to the use of AMPs (Table 2 [43,44,45,46,47,48,49], Figure 4), which can be produced by various living organisms and can also be obtained semi-synthetically or synthetically [109,110].
microorganisms-12-01650-t002_Table 2Table 2Inhibition of polymicrobial consortia by antimicrobial peptides (AMPs).ConsortiaForm and Site (Reason) of PresenceSuppressive CompoundsEffects*C. albicans*/*P. aeruginosa*, *Acinetobacter baumannii*, *E. coli*, *K. pneumoniae* [43]Biofilm, medical devices (catheters, endotracheal tubes, contact lenses);infection of the oral cavity, cystic fibrosis lungs, wounds, abdominal cavity and urinary tractSynthetic mimic AMP–ceragenins (CSA-13, CSA-90) synthesized from a cholic acid scaffold technique) or natural AMP: magainin, cecropin A and LL-37 (100 µg/mL)CSA-13, CSA-90 provided 100-fold decrease of cell concentrations in multispecies biofilms. All other AMPs were ineffective except magainin against only *Candida* and *E. coli* with log reduction of cell concentrations*C. albicans*/*Achromobacter xylosoxidans* or *Stenotrophomonas maltophilia* [44]Biofilm, cystic fibrosisSynthetic AMP (mimic of myxinidin)—WMR (4–20 μM)40–50% disruption of biofilms *C. albicans*/*S. aureus* [45]Biofilm, catheter-related bloodstream infectionsSynthetic AMP (mimic of arginine, and alanine) guanylated polymethacrylates(128 mg/mL)99% death of *S. aureus* and 82% death of *C. albicans**C. albicans*/*K.pneumoniae* [46]Biofilm, intravascular or urinary catheters, oral infectionsSynthetic AMP (mimic of glycoprotein H of herpes simplex virus type 1) gH625-M (50 µM)50% inhibition of cell growth and eradication of polymicrobial biofilm **AMP in suppressive combinations**Carbapenem-resistant *Pseudomonas aeruginosa* and *Candida* sp. [47]Biofilm, wound infection, chronic lung disease, pneumonia associated with mechanical ventilation, and bloodstream infectionsNatural AMP polymyxin B (2 μg/mL) with caspofungin (32–64 μg/mL) with DMSODecrease of the cell concentration to 1–2 log CFU/mL and loss of total biofilm biomass*Aspergillus fumigatus*/carbapenem-resistant *P. aeruginosa* [48]Biofilm, cystic fibrosisCombination of natural AMP—polymyxin B (2 μg/mL) with caspofungin (64 μg/mL) and DMSOAlteration of hyphae morphology and death of bacterial cells with a decrease of bacterial cells to 2.0–3.0 log CFU/mL*Fusarium oxysporum*/*C. albicans* [49]Biofilm, mycosesSynthetic mimic AMP (modified myxinidin)—WMR (12.5 μM) with fluconazole (30 μM)60% decrease of biomass 


The mechanism of action of AMPs primarily consists of disrupting the integrity and functions of plasma membranes and cell walls of various microorganisms, which is why the development of resistance to AMPs does not occur in most microorganisms. Some AMPs can induce intracellular production of reactive oxygen species (ROS), which destroy lipids, proteins, and nucleic acids through their active oxidation [109] or inhibit the synthesis of mycotoxins by fungi, thus reducing the level of their own protection [109,111].

Semi-synthetic AMPs are chemically modified natural peptides, including phosphorylation, cyclization, halogenation, etc., while synthetic AMPs [43,44,45,46] are biomimetics of natural peptides, which are obtained by chemical synthesis using computer-aided molecular design methods aimed at increasing the toxicity of the resulting compounds towards pathogens [110,112,113]. Thus, two peptides (PepGAT and PepKAA) with an antimicrobial effect against *Candida* cells, including their biofilms, were synthesized and characterized [114].

AMPs with a broad spectrum of action or their combination with other antimicrobial drugs are required to influence mixed polymicrobial consortia. Thus, it was shown that natural AMPs (Magainin, Cecropin A, LL-37) at a concentration of 100 µg/mL were ineffective against mixed biofilms formed by *C. albicans* and gram-negative bacterial cells *P. aeruginosa*, *A. baumannii*, *E. coli* and *K. pneumoniae* [43].

A new group of synthetic AMP analogues, ceragenins CSA-13 and CSA-90, was found to be highly effective against mono- and polymicrobial biofilms [43].

In another work, a stable synthetic peptide WMR-4 was obtained, and its ability to inhibit and destroy two-species biofilms formed by *C. albicans*/*S. maltophilia* or *C. albicans*/*A. xylosoxydans* was investigated (Table 2) [44]. At a concentration of 20 μM, this WMR-4 peptide inhibited biofilm formation by 75%, but its activity against mature biofilms decreased to 40–50%.

Today, membrane-tropic peptides, which have a high ability to destroy polymicrobial biofilms, are of particular interest among the AMPs being studied. These AMPs include the membrane-tropic peptide gH625-M, which was used against a mixed biofilm of *C. albicans*/*K. pneumoniae* [46]. At a concentration of 50 μM, gH625-M was able to inhibit in vitro the formation of polymicrobial biofilm and also destroy it with an efficiency of about 50%. The primary destructive effect of gH625-M is directed at the exopolymer matrix of the biofilm, and then it penetrates through cell membranes into cells and causes local and temporary destabilization of intracellular membranes. As a result, there is a decrease in the adhesion and aggregation of yeast cells, which leads to a decrease in the strength, integrity and rate of biofilm maturation.

Other synthetic biomimetics of natural AMPs, such as guanylated polymethacrylates (128 mg/mL), suppressed the growth of *C. albicans* yeast cells by 82.2% in mixed biofilms [45]. However, it was noted that the antimicrobial activity of guanylated polymethacrylates against biofilms is 2–4 times lower compared to planktonic cells.

Thus, there is a great interest in synthetic AMPs, but it turned out that they, like natural versions of AMPs, are less effective in their suppressive effect on biofilms compared to cells outside the QS. This dictates the need to search and study their combinations with other antimicrobial agents. Thus, the combined application of polymyxin B (2 μg/mL) with the antifungal drug caspofungin (32–64 μg/mL) against yeast-bacterial biofilms of *Candida* sp./*P. aeruginosa* was demonstrated. Moreover, this combination most significantly reduced the number of *C. glabrata* yeast in the biofilm (reduction of cells by 1–2 log CFU/mL). However, *P. aeruginosa* cells in the biofilm turned out to be less susceptible, and a decrease in the number of bacteria was observed only in the consortium with *C. tropicalis* [47].

A similar combination of caspofungin (64 μg/mL) and polymyxin (2 μg/mL) was used against *Aspergillus fumigatus*/*P. aeruginosa* biofilm [48]. As a result, a change in the morphology of the hyphae of the filamentous fungus was revealed, and the biomass of the mixed biofilm (>50%) and the viability of bacterial cells in it were significantly reduced.

The activity of such a synthetic AMP as WMR (12.5 μM), which is a modified myxinidin, was tested in combination with fluconazole (30 μM) in the destruction of *C. albicans*/*Fusarium oxysporum* biofilm. The use of this combination significantly reduced the cell viability of both filamentous fungi and yeast (up to 60%) in mature biofilms, and these effects coincided with a decrease in the expression of genes responsible for QS and biofilm formation in both microorganisms [49].

Thus, it should be noted that for more effective suppression of mixed polymicrobial consortia in the state of biofilms, combinations of synthetic or natural AMPs with other antimicrobial drugs are necessary since cells in biofilms are less sensitive to the action of AMPs. Although AMPs themselves do not simply stop the growth of cells in biofilms, they actively contribute to their death.

### 2.3. Inhibition of Polymicrobial Consortia by Commercial Antimicrobials and Organic Solvents

Currently, substances with proven antimicrobial activity are used to suppress microbial consortia containing yeasts or fungi (Table 3) [50,51,52,53,54,55,56,57,58,59,60,61,62,63,64,65,66,67], and azoles are the most popular among them [51,52,60,61]. Interestingly, classical antibiotics, used against bacteria with the hope that bacterial cells, as part of mixed consortia, will be sensitive to such treatment, suppress the growth of some bacteria only being used in high concentrations [50], but at the same time can be effective against filamentous fungi [62].
microorganisms-12-01650-t003_Table 3Table 3Inhibition of polymicrobial consortia by commercial antimicrobials and solvents.ConsortiaForm and Site (Reason) of PresenceSuppressive CompoundsEffectsAirborne bacteria (*Staphylococcus*, *Bacillus*) and fungi (*Cladosporioides*, *Rhodotorula* sp.) [50] *Bacterial-fungal aerosols, wastewater treatment plantsAmoxycillin, ampicillin, ceftazidime, cefalotin, cefuroxime sodium, nalidixic acid, amikacin, doxycycline, erythromycin, gentamicin, kanamycin, neomycin, streptomycin, tobramycin, tetracycline, trimethoprim, rifampicin, chloramphenicol, nitrofurantoin and novobiocin (5–200 μg)12% decrease in concentrations of bacterial cells was reached, and *Bacillus mycoides* demonstrated the highest resistance to antibiotics*C. albicans*/*Klebsiella pneumoniae* or *S. aureus* [51]Biofilms, oral infections, respiratory diseasesTetrazole oteseconazole VT-1161 (2.0 µg/mL) with DMSOInhibition of fungal CYP51 and 90% eradication of biofilms *F. solani*/*S. aureus* or*S. epidermidis* [52]Biofilms,human cadaveric corneaAmikacin, gentamicin, tobramycin, ampicillin, cephalosporins, cefuroxime, ceftriaxone, cefepine, cefazolin, gatifloxacin, moxifloxacin, ciprofloxacin, ofloxacin, chloramphenicol, azithromycin, metronidazole, clindamycin, lincomycin, monocycline (12–1024 µg/mL) with ethanolMinimum biofilm eradication concentration (MBEC) (µg/mL) for:biofilm *S. aureus*/*F. solani*chloramphenicol −128monocycline −128ampicillin −256other antibiotics ≥480biofilm *S. epidermidis*/*F. solani*ciprofloxacin −64tobramycin −256chloramphenicol −256other antibiotics ≥512*C. parapsilosis*/*S. aureus* [53]Biofilm, bloodstream infectionsResolvin D1 (25 µg/mL) with ethanol80% suppression of expression of the genes, participating in the formation of biofilms *C. albicans*/*S. aureus* [54]Biofilm, periodontitis, cystic fibrosis, stomatitis, urinary tract, burn wound infectionsQuaternary ammonium compound based on terbinafine and pyridoxine (KFU-127) (400–800 mg/L) with methanolInhibition of growth of both bacterial and fungal cells inside the biofilm*C. albicans*/*S. epidermidis* [55]Biofilm, catheters50% solution EDTA (Ethylenediaminetetraacetic acid) in ethanolDramatically reduced mass of biofilms*C. albicans*/*S. aureus*/*P. aeruginosa* [56]Biofilms, wound infectionsG21-cholic acid derived amphiphile (8 μg/mL) with methanolDecrease concentration of cells in biofilms from 9–10 log CFU/mL down to 4–5 log CFU/mL*C. albican* or *C. auris*/*S. aureus* [57]Biofilms, mycotic infections100% surface disinfectant NSSD99.9% death of cells*C. parapsilosis*/*S. aureus* [58]Biofilm, nosocomial infectionsOrthophthalaldehyde (0.55% solution)Bacteriostatic and fungicidal activity decreases the concentration of yeasts from 7 log CFU/mL down to 2 log CFU/mL and bacteria from 8 log CFU/mL down to 4 log CFU/mL*Alternarla*, *Cladosporium*, *Fusarium*, *Pénicillium*, *Phoma*, *Trichoderma* and*Ulocladium genera* [59] *Co-colonization, wooden constructionsCommercial wood preservatives: “Borolitas”, “WT Sodium Hypochlorite; “Anti-mould liquid”, “Boramon”, “Arlitas”, “Complete Wood Treatment”*Alternaria* and *Fusarium* were the most tolerant to wood preservatives, whereas growth of *Penicillium* cells was suppressed41 fungal isolates including *Aspergillus* fumigatus, *Fusarium* oxysporum, and *Candida* sp. [60] *Consortia,wastewater treatment plantsFluconazole, ketoconazole, itraconazole, and voriconazole (0.06–64 μg/mL)MIC ** values for all tested antifungals against *Candida krusei* and *A. fumigatus* ≤ 1 μg/mL except fluconazole (64 μg/mL). MIC values for *F. oxysporum* were ≥4 μg/mL*A. fumigatus*, *A. lentulus* [61]Mixed infection, cystic fibrosisVoriconazole (10 mg/kg)The invasive growth of *A. lentulus* was observed in mixed infections after antifungal treatmentMicrobial communities (*Firmicutes*, *Proteobacteria*, *Neocallimastigomycota*, *Basidiomycota*, *Bacteroidetes*, *Ascomycota* etc.) [62]Consortia, aerobic compostingTetracycline hydrochloride (50–300 mg/kg)Low concentrations of antibiotic promote *Chytridiomycota* growth, while high concentrations inhibit fungal activity*Acinetobacter* sp., *E. coli*,*Pseudomonas* sp., *Staphylococcus* sp., *Desulfovibrio* sp., *Clostridium* sp. *Penicillium* sp., *Fusarium* sp., *Cladosporium* sp., *Rhizopus* sp., *Aspergillus* sp., *Candida* sp. [63]Consortia,biofilms,oil and gas industry3.3′-metylenebis[5-methyloksazolidine] (MBO) (2%)100% death of cells in biofilmsAnaerobic sludge [64]Consortia,process of biogas productionDibenzothiophene sulfone (0.45 mM)30% decrease in the metabolic activity of cells **Suppressive combinations***C. albicans*/*S. aureus*/*E.coli* [65]Biofilms,medical devicesCombination of moxifloxacin (6 mg/L) with caspofungin (12.5 mg/L) or meropenem (30 mg/L) with caspofungin (12.5 mg/L) in DMSOBoth combinations were able to reduce the cells in biofilms, and a *C.albicans* cells were dead after incubation with meropenem-caspofungin*C. albicans*/*S. aureus* [66]Biofilms,medical devices, skin, mucosal, and bloodstream infections Combination of chalcone-based derivative (53 μM) with polycyclic anthracene-maleimide (4 μM) in DMSOUp to 64% biomass reduction in biofilm *A. fumigatus*/*Stenotrophomonas**maltophilia* [67]Biofilms, cystic fibrosisAmphotericin B (64 μg/mL) with levofloxacin (4 μg/mL) with DMSOBiomass inhibition −90%*S. maltophilia* increased the antifungal susceptibility of *A. fumigatus* to amphotericin B, whereas *A. fumigatus* protected *S. maltophilia* from levofloxacin* Microorganisms were isolated from consortia and the effects of antimicrobials were assessed on individual cultures. ** MIC—minimal inhibiting concentration.


The expectation that the use of antibacterial agents will be effective in relation to consortia containing bacteria and fungi is based on the fact that metabolic and trophic connections between microorganisms in polymicrobial biosystems will be disrupted as a result of the death of bacteria. It is known that bacteria and fungi in such consortia can be in mutually beneficial cooperation or in a state of balanced competition. For example, it has been established that the presence of the yeast *Exophiala dermatitidis* in mixed consortia can lead to both a sharp increase in the growth of their biofilms with bacteria [11] and a decrease in the intensity of the formation of bacterial exopolysaccharides in the case of consortia with the fungi *Aspergillus niger* [27].

Experiments showed that mixed polymicrobial biofilms of *S. aureus*, *S. epidermidis* and *F. solani* were less sensitive to various antibacterial and antifungal drugs compared to corresponding planktonic cells when 18 antibiotics of different classes and 6 antifungals were investigated [52]. An increase in the minimum biofilm eradication concentration (MBEC) was confirmed. For biofilms of *S. aureus*/*F. solani*, the lowest MBEC values were found in the case of chloramphenicol and monocycline (128 µg/mL), while in the case of ampicillin it was two times higher. For biofilms of *S. epidermidis*/*F. solani*, the best results were shown by ciprofloxacin (64 µg/mL), while tobramycin and chloramphenicol began to act only at a concentration four times higher than this. Poor penetration of antibiotics and antifungals through the matrix of the formed biofilm was explained by the authors of the study as a reason for the increase in MBEC values.

Tetrazole otseconazole (VT-1161), as a member of a new generation of fungal cytochrome P450 family 51 (CYP51) inhibitors, has been evaluated in its action on *C. albicans*/*K. pneumoniae*/*S. aureus* biofilms. At a concentration of 0.5 μg/mL, it inhibited the formation of biofilm, and when the concentration increased fourfold, it caused the eradication of mature biofilm by 90% [61]. A decrease in the adhesion of *Candida* cells to adult prostate epithelial cells PNT1A when treated with VT-1161 was also shown [51].

It has been shown that ω3-polyunsaturated fatty acids (eicosapentaenoic and docosahexaenoic) have antimicrobial properties against a wide range of microorganisms due to the provoked by them oxidative stress, leading to cell apoptosis [53]. In this regard, the possibility of using resolvin D1 (RvD1) synthesized from docosahexaenoic acid as an antimicrobial agent against *S. aureus*/*C. parapsilosis* biofilms formed on polystyrene and silicone was investigated. RvD1 ensured 80% cell death in the biofilm by stimulating the accumulation of ROS and suppressing the expression of genes involved in biofilm formation (bap, icaD and icaA for *S. aureus* and ALS3 adhesins of *C. parapsilosis*) [53].

Quaternary ammonium compounds (benzalkonium chloride, miramistin and cetylpyridinium chloride) are widely used as antimicrobial agents, but they are ineffective against polymicrobial biofilms. Therefore, new variants of quaternary ammonium derivatives are being developed, in particular, terbinafine based on pyridoxine, which has an antimicrobial effect [54]. Compound KFU-127 has been shown to provide a 3-log drop of CFU/mL in mixed *S. aureus*/*C. albicans* biofilms. The mechanism of antimicrobial action of KFU-127 is based on damaging the membrane integrity and on the impact on pyridoxal-dependent enzymes. However, the high concentration of KFU-127 (800 mg/mL) that produces this effect was found to be highly toxic to mammalian cells.

A cholic acid-derived gemini amphiphile (G21) was developed, which interacted with the lipid components of bacterial and fungal membranes, leading to their damage in polymicrobial biofilms [56]. A rather low concentration of G21 (8 µg/mL) led to cell death and destruction of mixed *S. aureus*/*C. albicans* or *P. aeruginosa*/*C. albicans* and *S. aureus*/*P.aeruginosa*/*C. albicans* biofilms. Although the use of G21 did not lead to the development of resistance in bacteria and fungi, the substance G21 turned out to be as highly toxic as KFU-127.

Chlorhexidine, actively used today as an effective antiseptic, does not act against mature yeast biofilms. Therefore, a new disinfectant, NSSD, was developed and shown to be highly effective against polymicrobial biofilms consisting of either *C. auris* or C. *albicans*/*S. aureus* (99% destruction of yeast and bacteria cells) [57]. But to achieve such results, its use in 100% concentration is required. Interestingly, a 50% EDTA solution in ethanol used to inhibit biofilm growth of *C. albicans*/*S. epidermidis* also led to a decrease in the rate of biofilm formation and size, but the concentration that provided this result was also too high, as in the case of NSSD [55].

Eleven commercial disinfectants were tested at varying concentrations and durations of exposure on *C. parapsilosis*/*S. aureus* biofilms. Among the disinfectants studied, the most effective was orthophthalaldehyde, applied at a concentration of 0.55% for 10 min. The reduction in yeast cell concentration was almost 4 log10 CFU/mL. However, this compound exhibited only bacteriostatic properties against the cells of pathogen *S. aureus*, in contrast to its action on yeast [58].

The suppression of natural mixed consortia presents a special practical challenge, which always begins with the choice of the substance itself, which, first of all, should not be highly toxic. The next criterion for selecting such a substance should be the effect of its use in a relatively low concentration and the possibility of degradation of its residual concentrations under environmental conditions after achieving the effect. An example of such a substance is dibenzothiophene sulfone (0.45 mM), which, as it turned out, can provide a 30% reduction in the metabolic activity of cells as part of a methanogenic consortium, which includes not only multiple representatives of bacteria and fungi but also archaea [64]. Under the influence of sulfate-reducing bacteria, such sulfone slowly turns into sulfide, and the rate of accumulation of unwanted biogas is significantly reduced.

As in the case of AMPs, combinations of antimicrobial agents with other compounds are currently attracting much attention from researchers when working with mixed biofilms. Thus, a combination of a chalcone derivative (53 μM) and a polycyclic anthracene-maleimide adduct (4 μM) was tested against *C. albicans*/*S. aureus* biofilms [66]. It has been shown that treatment with these drugs completely changes the architecture of the mixed biofilm; namely, there is a decrease in cellular aggregates along with a significant decrease in the number of hyphal forms of *C. albicans*. The concentration of *C. albicans* cells does not decrease significantly, but the transition of yeast to the hyphal form is noticeably impaired in proportion to the concentration of the chalcone derivative. At the same time, a decrease in the number of cells of the bacterial pathogen *S. aureus* is observed. The proposed mechanism of the antimicrobial action of the combination used is the inhibition of the cellular enzymes 1,3-β-glucan synthase and chitin synthase. Interestingly, a decrease in the concentration of nucleic acids inside cells was noted, possibly due to inhibition of mRNA synthesis or degradation of DNA and RNA.

Combinations of broad-spectrum antibiotics (moxifloxacin, meropenem) and an antifungal agent (caspofungin) have been tested against mixed bacterial and yeast biofilms (*S. aureus*, *E. coli*, and *C. albicans*) in vitro [65]. It was shown that the combination of meropenem (30 mg/L) and caspofungin (12.5 mg/L) was most effective against three-component biofilm, leading to the death of the maximum number of yeast cells. At the same time, a decrease in the content of exopolysaccharides in the biofilm was noted. It is interesting to note that moxifloxacin turned out to be the most ineffective in its action against the biofilms studied. Moxifloxacin and meropenem did not prevent the increase in the number of yeast cells in biofilms after bacterial cell death.

The combination of amphotericin B (64 μg/mL) with levofloxacin (4 μg/mL) was effective against the polymicrobial biofilm of *A. fumigatus*/*Stenotrophomonas maltophilia*, inhibiting by 90% the growth of both fungi and bacteria. The sensitivity of fungi in the biofilm to individually applied amphotericin B increased in comparison with planktonic cells of the fungus, while the susceptibility of bacteria did not change. In the case of individual use of levofloxacin, the susceptibility of bacteria to it decreased in biofilms and depended on the biomass of the fungi present [67].

Thus, the responses of cells in mixed biofilms to individual antimicrobial agents may vary greatly in comparison with planktonic cells or monogenic biofilms. The result of using combinations of antimicrobial agents based on existing knowledge about the reaction of microorganisms to them is actually theoretically unpredictable and requires individual verification. At the same time, today, the tendency towards a decrease in the effectiveness of used commercial antibacterial drugs and fungicides, especially against mixed polymicrobial consortia, has become quite obvious. To maintain the possibility of using available blockbuster antibiotics against biological systems that are heterogeneous in composition, it is necessary to combine the use of different antimicrobial agents in certain combinations. These combinations must be sought experimentally, and it is important to control the toxicity of the created combined solutions.

### 2.4. Heavy Metals and Nanoparticles (NPs) against Mixed Consortia of Microorganisms

It is well known that heavy metal ions inhibit the enzymes of microorganisms and, thus, suppress their metabolic activity [25,68,69]. However, today, metal NPs attract researchers’ attention (Table 4; Figure 5) [26,70,71,72,73,74,75,76,77,78] since they can interact with cell membranes and nucleic acids, contributing to the generation of ROS, which leads to their antimicrobial effect. By varying the shape, size, charge and hydrophilicity of metal NPs, it is possible to control their delivery to biofilms [115].
microorganisms-12-01650-t004_Table 4Table 4Metal ions and NPs in the inhibition of polymicrobial consortia.ConsortiaForm and Site (Reason) of PresenceSuppressive CompoundsEffectsSuperficial deposits-bacteria (*Bacteroidetes*, *Proteobacteria*, *Actinobacteria*) and dark-colored fungi (*Alternaria alternata*, *Cladosporium cladosporioides*, *Coniosporium* sp., *Phoma herbarum*, *Aureobasidium pullulans*) and *A. niger* [25]Consortia, stone monumentsFe, Mn, Zn, Cu, Pb, and Cd (8.5–32,280.6 μg/g)Resistance of consortia to the action of heavy metals*Chlorella vulgaris*/*Aspergillus oryzae* [68]Consortium, wastewater treatment plantsCu (II) (0.7–1.0 mg/L)Decrease of metabolic activity of cells*Aspergillus lentulus*, *A. terreus* and *Rhizopus oryzae* [69]Consortium,wastewater treatment Cr(VI) and Cu(II) (75 mg/L)No inhibition*C. albicans*/*S. aureus* [70]Biofilm, human infectionsAg NP (32 μg/mL)Eradication of biofilm*Rhodotorula* sp., *Debaryomyces hansenii* and *Hanseniaspora valbyensis* [71]Consortium, bioremediation processesNP of ZnO—(3.0 g/L)Decrease of metabolic activity of cells*Diversispora versiformis*, *Funneliformis dimorphicus* and *Glomus indicum* [72]Symbiosis,agriculture, arbuscular mycorrhizal fungiNP of TiO_2_—(100 mg/plant)Inhibition of fungal growth due to the binding of TiO_2_ with plant roots or increases in internal concentration of TiO_2_ in root tissue*Glomus versiforme* and *G. caledonium* [73] *Symbiosis, agriculture, arbuscular mycorrhizal fungiNP of ZnO—(800 mg/kg)Inhibition of fungal colonization of plant roots**Combinations with metal NP**Lithobiotic microbial community (bacteria, microscopic fungi, algae, and lichens) [26]Consortium,stone monuments Sol-gel-derived epoxysiloxane coatings with 0.3–0.5 wt% NP of TiO_2_ P25 in combination with nanodiamond powderInhibition of the micromycetes growth*C. albicans*/*S. aureus or Streptococcus mutans* [74]Biofilms,oral cavity infection, otitis, chronic lung infection, burn wounds, urinary tract infectionFucoidan-AuNP (Fu-Au NP) (2048 µg/mL)Decrease of cell concentration down to 2.4–5.8 log CFU/mL as a result of inhibition of expression of genes involved in the biofilm formation, altering of membrane penetration and ROS generation*C. albicans*/*S. aureus* [75]Biofilm,skin, mucosal, and bloodstream infectionsβ-caryophyllene-Au NP (1024 μg/mL)Decrease of *S. aureus* and *C. albicans* cells in the mixed biofilm by 1.4 log CFU/mL and 2.1 log CFU/mL, respectively.*C. albicans*/*S. aureus* [76]Biofilm,nosocomial infectionPhloroglucinol-Au NP (1024 μg/mL)Decrease of cells in mixed-biofilm down to 2.8 log CFU/mL.*C. albicans*/*S. aureus* [77]Biofilm,vaginitisAg NP and L-carnitine (1000 ppm)90% inhibition of biofilm growth*C. albicans*/*S. mutans*/*E. faecalis* [78]Biofilm,oral cavity infectionsSulfonated lignin-with Pd NP (SLS-Pd) (1.65 mg/mL) in combination with near-infrared (NIR) irradiation (808 nm, 1W cm^−2^)50% eradication of biofilm* Microorganisms were isolated from consortia and the effects of antimicrobials were assessed on individual cultures.


The use of NPs in combination with antimicrobial agents seems to be more effective and successful in suppressing the development of biofilms (Table 4 [26,74,75,76,77,78]).

Gold NPs (AuNPs) have been shown to inhibit biofilm growth [74,75,76]. AuNPs, in combination with the use of near-infrared light, aggregate and absorb light with a red shift, which leads to a sharp increase in localized heat at the site of the introduction of AuNPs into the biofilm.

It has been found that the antibacterial properties of AuNPs can be enhanced by combining them with fucoidan (Fu-AuNPs) [76]. Such particles showed high efficiency in inhibiting yeast-bacterial biofilms (*C. albicans*/*S. aureus* or *Streptococcus mutans*). It is worth noting that Fu-AuNPs inhibited monospecies biofilms more effectively than polymicrobial ones. The mechanism of biofilm eradication under the influence of Fu-AuNPs included simultaneous inhibition of the expression of genes associated with biofilm formation, changes in membrane permeability and the generation of ROS.

In another study, AuNPs were synthesized using β-caryophyllene (β-c-AuNPs) and applied against *C. albicans*/*S. aureus* mixed biofilm. Similarly, many studies of metal NPs have shown inhibition of biofilms, depending on the concentration of NPs.

In one another work, a 1024 μg/mL solution of NPs led to a decrease in the concentration of cells in biofilms by 1.4–2 orders of magnitude [75]. The combination of a solution of the same concentration of AuNP with phloroglucine (PG-AuNP) against biofilms of a similar composition (*C. albicans*/*S. aureus*) reduced the cell number by almost three orders of magnitude [76]. Thus, the combination of metal NPs with the choice of combination partners turned out to be very important.

Silver NPs (AgNPs) are also widely used as antimicrobial agents [70]. Thus, treatment of *C. albicans*/*S. aureus* biofilm with these NPs (32 μg/mL) showed that they are adsorbed on the cell surface with subsequent penetration into the cells, generate ROS, attack the respiratory chain and disrupt cell division, which ultimately leads to their death. After treatment with such NPs, biofilms were heterogeneous with changes in cell morphology [70].

Despite the widespread use of AgNPs, toxicological studies have confirmed their toxicity to liver cells, alveolar epithelium and cells of the nervous and endocrine systems due to the generation of ROS. In this regard, there was an interesting study in which AgNPs were combined with L-carnitine; L-carnitine effectively reduced the toxic effects of metal NPs, and at the same time AgNPs with L-carnitine suppressed the growth of yeast-bacterial biofilms *C. albicans*/*S. aureus* in the treatment of vaginitis in mice [77].

Another effective modern strategy for suppressing consortia that are heterogeneous in composition is the combination of metal NPs with mimetics of oxidative enzymes. Thus, palladium NPs together with sulfonated lignin (SLS-Pd), exhibiting oxidase-like properties, demonstrated high efficiency in destroying biofilms formed by *S. mutans*, *E. faecalis* and *C. albicans* cells in vitro [78]. SLS-Pd exerted a selective photothermal lethal effect on *C. albicans* yeast cells inside polymicrobial biofilms. Thus, at pH 4.5 and near-infrared (NIR) light irradiation, eradication of the polymicrobial biofilm occurred with a 40% decrease in its relative biovolume. At the same time, the oxidase-like activity of SLS-Pd was more pronounced against bacterial cells (*S. mutans* and *E. faecalis*).

However, the main problem with the use of metal NPs is their toxicity [116]. In addition, recent studies have shown the adaptation of microbial cells to oxidative stress caused by metal NPs. Due to these disadvantages, the use of metal NPs in clinical practice is limited [117]. Since combinations of NPs and antimicrobial agents can potentially reduce the concentration of used toxic substances due to a synergistic suppressive effect, this direction can be considered promising for the further development of drugs against mixed consortia involving fungal cells.

### 2.5. Physicochemical Methods for Suppression of Polymicrobial Consortia

Since the use of near-infrared (NIR) light irradiation was mentioned above [78], the question of the feasibility and effectiveness of using various physical and chemical methods, in particular, photodynamic effects on biofilms formed by polymicrobial consortia, is obvious. Photodynamic treatment involves the use of photosensitizers, which, when irradiated with light, give up their energy or electrons to molecular oxygen, forming ROS that cause damage to microbial cells (Table 5 [79,80,81,82,83,84]).
microorganisms-12-01650-t005_Table 5Table 5Inhibition of polymicrobial consortia by ions using physicochemical methods.ConsortiaForm and Site (Reason) of PresenceSuppressive CompoundsEffects*C. albicans*/*E. faecalis* [79]Biofilm, endodontic infectionsZn(II)chlorin e_6_ methyl ester (0.1 mg/L) with DMSO and light activation for 60–90 s60% Removal of biomass from biofilm*C. albicans*/*P. aeruginosa* [80]Biofilm, burn wound, chronic lung infectionsBlue Light irradiation (405 nm), 90 minInactivation of *P. aeruginosa* 2.58-log10 CFU/mL and *C. albicans* 1.71-log10 CFU/mLSoil bacteria and fungi [81]Consortium, soil pH 4.0–8.0Negative interactions between bacteria and fungi at alkaline pH Activated sludge (*Trichoderma*, *Cutaneotrichosporon*, *Nitrosomonas*, *Nitrospira*, *Dechoromonas*, *Rhodanobacter*) [82]Consortium, wastewater treatment plantspH 5.5Growth of fungi and inhibition of nitrogen removal by bacterial cells*Citrobacter freundii*/*Sphingobacterium multivorum*/*Coniochaeta* sp. [83] *Consortium, degradation of wheat strawpH 5.2–7.2 or shaking speed (60–180 rpm)Suppression of fungal growth at pH 5.2–6.2 and 180 rpm in the presence of bacterial cells*Aureobasidium* sp., *Cladosporium* sp., *Penicillium* sp. [84]Biofilms, consortia, water plantsFiltration with ozonation or chlorination90% decrease in fungal cells * Microorganisms were isolated from consortia, and the effects of antimicrobials were assessed on individual cultures.


For example, the effectiveness of antimicrobial photodynamic therapy using red light-activated Zn(II)chlorin e6 methyl ester (Zn(II)e6Me) against mixed biofilms of *E. faecalis* and *C. albicans* was evaluated. After activation with light for 60–90 s Zn(II)e6Me led to a 60% decrease in cell biomass in the biofilm. The disruption of proteins and lipids in the membranes of *E. faecalis* cells and the leakage of cellular contents were confirmed. A number of intracellular changes were noted in *C. albicans* cells (disturbance in the organization of the cytoplasmic membrane, morphology of vacuoles, and damage to other organelles). However, in the dark, a decrease in the activity of Zn(II)e6Me was observed [79].

It is known that blue light (aBL) in the spectrum from 400 to 470 nm has an antibacterial effect without the additional use of photosensitizers. The proposed mechanism of the antimicrobial effect of irradiation with such light involves the accumulation of photoactivated metal-free porphyrins, such as uroporphyrin and coproporphyrin, in microbial cells. It is known that porphyrins, after irradiation with light, transform into a triplet state, in which singlet oxygen is generated, which reacts with a wide range of cellular macromolecules and damages them (proteins, lipids, DNA and RNA). In addition, the possible mutually damaging effect of cells on their partners in biofilms due to their own damage is discussed. For example, the effectiveness of blue light (aBL; 405 nm) for inactivating biofilms (*C. albicans*/*P. aeruginosa*) was studied [93]. Inactivation of cells with a decrease in their concentration was observed (inactivation of *P. aeruginosa* 2.58-log10 CFU/mL and *C. albicans* 1.71-log10 CFU/mL). It was found that the decrease in yeast concentration is associated not only with their treatment with light but also with the death of *P. aeruginosa* cells in mixed biofilms, which form their biofilm on the hyphae of *C. albicans*. With the active death of bacterial cells, yeast hyphae were also damaged. 

Today, the simplest physicochemical way to suppress consortia containing filamentous fungi appears to be by varying the pH of the environment. At low pH values, filamentous fungi begin to predominate in the community, and at neutral or alkaline pH values, bacteria begin to predominate [81,82,83]. A similar approach to suppressing consortia is used, as a rule, in the case when the harm is caused not by the microorganisms of the consortium as a whole but by individual metabolites of bacteria or fungi.

A common method of water purification, combining chemical (ozonation, chlorination) and physical (filtration) approaches, allows achieving 90% removal of biofilms [84]. The use of combinations of different methods makes it possible to level out the disadvantages of each individually but requires careful monitoring of the toxicity of the resulting aquatic media.

### 2.6. Microbial Cells and Their Metabolites as Means of Suppression of Polymicrobial Consortia

#### 2.6.1. Enzymes as Antimicrobial Agents

One of the directions in developing a strategy to combat polymicrobial infections is focused on the use of various metabolites produced by microorganisms, among which enzymes such as oxidoreductases and hydrolases attract special attention (Table 6 [23,85,86,87,88,89,90,91,92], Figure 6).
microorganisms-12-01650-t006_Table 6Table 6Enzymes against polymicrobial consortia.ConsortiaForm and Site (Reason) of PresenceSuppressive CompoundsEffects*C. albicans*/*S. aureus* or *E.coli* or *P. aeruginosa* or *K. pneumoniae* [85]Biofilms, bloodstream infectionsBovhyaluronidase azoximer Longidaza^®^ (750 IU)30–40% decrease of biomass in biofilm*C. albicans*/*S. aureus* [86]Biofilm, bloodstream infectionsCombination of cellobiose dehydrogenase and deoxyribonuclease I (co-immobilization on chitosan NP (1–2 mM))Penetration through the biofilm matrix and 90.5%inhibition of the biofilm formation *S. cerevisiae*/*Flavimonas oryzihabitans*, *Lactobacillus brevis*, *Leuconostoc mesenteroides* [87]Biofilms, dispense equipmentCombination of enzymes: α-amylase (10 U/m), β-glucuronidase (10 U/mL), glucose oxidase (10 U/mL), dextranase (1 U/mL), protease (10 U/mL) and pectinase (60 U/mL)Removal of *L. brevis* and *L.mesenteroides* cells from biofilms, but not of *S. cerevisiae* and *F. oryzihabitans**Fusarium* sp./*Alternaria* sp. [88]Symbiont of pathogens, banana fungal diseasesCombination of chitinase and β-1,3-glucanase from *Penicillium* sp. and *Bacillus* sp. 60% decrease in banana disease *Fusarium* spp., *Alternaria* sp., *Cladosporium* sp. [89]Fungal communities, olive tree twigsCombination of antibiotics and fungal cell wall degrading enzymes from *Pseudomonas savastanoi pv. savastanoi*Preventing fungal colonization and proliferation on the surface *Trichoderma* sp. [90] *Consortia,mushroom farms*Bacillus subtilis* (5 × 10^8^ CFU/mL) producing antibiotics, β-1,3-glucanase, chitinase, protease, lipase, amylase siderophores, pyrazine, 2, 3-dimethyl-5-(1-methylpropyl)] 48–52% inhibition of fungal growth *C. albicans*/*S. epidermidis* [91]Biofilm, catheter-associated urinary tract infectionDNase I or marine bacterial DNase from *Vibrio alginolyticus* (5 µg/mL) with biosurfactant from *Bacillus subtilis* (300 μg/mL)79–85% inhibition of biofilm formation due to inhibition of *C. albicans* hyphal appearance Grape must consortium [92]Consortia, wine makingRecombinant β-glucanase as a yeast killer toxin (LrKpkt) from *Tetrapisispora phaffi* (2 AU/mL)90% decrease in cell viabilityAnaerobic sludges [23]Consortia, landfillsHis_6_-OPH ** (5 mg/L) in combination with bacitracin (100 mg/L), potassium humate modified with naphthoquinone (CHP-NQ 5 g/L), and K_2_S_2_O_8_ (4 g/L) 34% decrease in biogas production and content of CH_4_ in it * Microorganisms were isolated from consortia, and the effects of antimicrobials were assessed on individual cultures. ** His_6_-OPH—hexahistidine-containing organophosphorus hydrolase.


Their main action is aimed at inhibiting the assembly of the cell wall, as well as at the hydrolysis of exopolysaccharides in the composition of biofilms that maintain the integrity of the established consortia. Enzymes can be used either individually [85] or in combination with other enzymes [86,87,88] and antimicrobial agents [23,89]. Since one of the components of the biofilm is extracellular DNA, deoxyribonuclease I was used in combination with cellobiose hydrogenase to suppress the formation of mixed biofilms (*C. albicans*/*S. aureus*) [86]. Enzymes can be co-immobilized on chitosan NPs to increase their stability. As a result of such stabilization, the biofilm inhibition efficiency reached 90%. Interestingly, chitosan NPs with only DNase did not affect the biofilm formation process. This should be especially taken into account since the cells of microorganisms remain viable and can again form a biofilm even if the integrity of the biofilm has been damaged.

The effectiveness of bovhyaluronidase azoximer (Longidase^®^) against *C. albicans* yeast biofilm mixed with *S. aureus*, *E. coli*, *P. aeruginosa* and *K. pneumoniae* bacteria was studied in vitro [85]. As a result, it was shown that treatment with Longidase^®^ (750 IU) reduced the biomass of yeast biofilms with various bacteria by 30–40%. The effectiveness of the enzyme depended on the composition of the extracellular matrix of biofilms. The enzyme itself did not affect the viability of the yeast *C. albicans*, and even in combination with fluconazole (256 μg/mL), complete cell death was not observed.

To inhibit a biofilm containing different types of bacteria *Flavimonas oryzihabitans*, *Lactobacillus brevis*, *Leuconostoc mesenteroides* and the yeast *Saccharomyces cerevisiae*, a mixture of enzymes was used: α-amylase, β-glucuronidase, glucose oxidase, dextranase, protease and pectinase [87]. Such combined treatment using an enzyme cocktail was most effective in reducing the number of viable cells in the biofilm, and this effect increased with increasing enzyme concentrations in the mixture. Yeast cells were less susceptible to the effects of enzymes compared to bacteria. This resulted in the enzymatic treatment being ineffective in the complete inhibition of the growth of all cells in the biofilm. For this, additional treatment with a chemical agent was necessary. In addition, the effectiveness of biofilm treatment with enzymes depended not only on the microbial cells present in it, but also on the surface on which the biofilm developed. Thus, the concentration of cells in the biofilm was significantly lower on stainless steel compared with dispense equipment.

A recombinant toxin Kpkt, which is a β-glucanase that induces ultra-structural modifications of the target cell wall usually produced by the yeast *Tetrapisispora phaffii*, was obtained using modified cells of the yeast *Komagataella phaffii*. This enzyme (toxin) has been used to inhibit other types of yeasts and bacteria present in grape must to partially replace SO_2_ [92]. Kpkt has been shown to be effective against the yeasts *D. bruxellensis*, *H. uvarum*, *K. apiculata*, *L. thermotolerans*, *S. cerevisiae*, *W. anomalus*. Among the bacteria *L. plantarum*, *E. coli*, *S. aureus* and *S. bongori*, all of which died when Kpkt was administered at a dose of 1 AU/mL, and for *L. rhamnosus* and *L. monocytogenes* cells, the effective dose was 2 AU/mL. When Kpkt (2 AU/mL) was introduced into grape musts, the concentration of yeast cells decreased by 90%. However, Kpkt was ineffective against filamentous fungi isolated from grape must (*Galactomyces* sp., *Fusarium* sp., *Aspergillus* sp., *Cladosporium* spp., *Rhizopus* spp., *Penicillium* sp.).

An interesting solution was to combine a biosurfactant (300 μg/mL) produced by *B. subtilis*, which is a lipopeptide consisting of pyrrolo[1,2-a]pyrazine-1,4-dione, hexahydro-3-(phenylmethyl)-octadecanoic acid methyl ester and hexadecanoic acid methyl ester, with DNase I or DNase from *Vibrio alginolyticus* (5 μg/mL). This combination was used against *C. albicans*/*S. epidermidis* biofilm. A significant inhibition of its growth was found (79–85%). The authors explained the resulting effect primarily by a decrease in surface tension, which prevented the adhesion of microbial cells to the surface [91].

The combination of hexahistidine-containing organophosphorus hydrolase, which hydrolyzes various lactone-containing quorum molecules of bacteria and fungi [23,118,119,120], with humate modified with naphthoquinone, the oxidizing agent K_2_S_2_O_8_ and AMP such as bacitracin, was successful having a suppressive effect on anaerobic consortia catalyzing methanogenic processes. It is this composition of four components that turned out to be more effective than each of its individual components in reducing the release of biogas and the proportion of methane in it [23]. Such suppressive combinations may be in demand at landfills.

From the presented material (Table 6), it is clear that the variety of enzymes and their targeted antimicrobial action makes it possible to reduce the formation of biofilms. Immobilization of enzymes can significantly increase their stability and functional efficiency. However, the main disadvantage of using enzymes to combat polymicrobial biofilms is their high cost. Further efforts are needed to develop active enzymes for large-scale use, including using computer modeling to find the most effective combinations of enzymes and anti-microbial agents to identify their possible interactions.

#### 2.6.2. Quorum Molecules in the Regulation of Polymicrobial Consortia

The mechanism for transmitting QS signals from cell to cell for the purpose of formation and regulation of biofilms is carried out by quorum molecules. In gram-negative bacterial cells, these are most often N-acyl homoserine lactones; in gram-positive bacterial cells, oligopeptides; in yeast, farnesol, tyrosol and phenylethanol; and in filamentous fungi, lactone-containing molecules and oxylipins [119,121]. In this regard, the use of quorum molecules to combat biofilms may be a promising strategy (Table 7 [27,93,94,95,96,97,98,99,100,101,102,103,104]).
microorganisms-12-01650-t007_Table 7Table 7Cell free supernatants, extracts with cell metabolites and cells of microorganisms against polymicrobial consortia.ConsortiaForm and Site (Reason) of PresenceSuppressive CompoundsEffects*C. tropicalis*/*C. krusei*/*C. parapsilosis* [93]Biofilms, oral infectionsCell free supernatants of *Lactobacillus gasseri* and *Lactobacillus rhamnosus* (30% *v*/*v*)56–67% decrease in biomass of surface-associated biofilm*C. albicans*/*S. aureus* [94]Biofilm, bloodstream infectionsDMSO extract of *Limosilactobacillus fermentum* metabolites (406 µg/mL)90% inhibition of biofilm formation and decrease in cell concentration*C. albicans*/*S. mutans* [95]Biofilm, cariesSupernatant of cultural broth after cultivation of *Lactobacillus plantarum* cells 91.5% and 43.7% decrease of *S. mutans* and *C. albicans* cells in the biofilm *C. albicans*/*C. tropicalis*, *S. salivarius*, *Rothia dentocariosa*, *S. epidermidis* [96]Biofilms, medical silicone devicesCell free supernatant of *Lactobacillus rhamnosus* (40% *v*/*v*)91% inhibition of cell adhesion to silicone and 58% decrease in cells viability*Aspergillus niger*/*Bacillus subtilis* [27]Co-culture, marble and limestone monumentsFungal and bacterial metabolites secreted by microorganisms to the brothSignificant effect on metabolic activity of cells*Pythium debaryanum*, *Fusarium oxysporum lycopersici*, *F. moniliforme* and *Rhizoctonia solani* [97]Consortium, disease of tomato seedlingsExtracts of compost containing *Anabaena variabilis*, *A. ocillarioides*, and vermiculite10–15% decrease in diseases of tomato seedlings*C. albicans*/*E. coli* [98]Biofilm, mucosal infections, vulvovaginal candidiasisFarnesol (600 µM) in DMSODegradation of biofilm*C. albicans*/*S. mutans* [99]Biofilm, caries*Streptococcus salivarius* LAB813 cells (10^7^ CFU/mL)90% death of cellsActivated sludge with *Zoophagus sp* [100] *Consortium, wastewater treatment plants*Clonostachys rosea* cells*C. rosea* cells penetrate to the interior of the *Zoophagus mycelium* and use the cytoplasm as a nutrition medium**Combinations***C. albicans*/*S. aureus* [101]Biofilm, cathetersRLmix_Arg—biosurfactants produced by *P. aeruginosa* with Pluronic F-127Decrease of cell viability*Rhodotorula mucilaginosa*/*Candida tropicalis*, *C. krusei*, *C. kefyr*, *Listeria monocytogenes*, *Salmonella enterica*, *Escherichia coli* [102]Biofilm, apple juice processing linesCombination of natamycin (10 µM) and farnesol (600 µM) in DMSOInhibition of biofilm growth due to decrease of filamentous formation by yeast cells, destabilization with defragmentation of 3D-structure of biofilm and concentration decrease of bacterial cells*C. albicans*/*Providencia stuartii*/*S. aureus* or*C. albicans*/*Acinetobacter baumannii*/*S. aureus* [103]Biofilms,nosocomial bloodstream infectionCombination of carvacrol (0.5 mg/mL) with farnesol (0.7 mg/mL) in DMSO75% inhibition of biofilm formation*C.albicans*/*S.aureus* [104]Biofilm, central venous catheters, urinary catheters, cardiovascular devicesCombination of farnesol (300 μM) with oxacillin (2 mg/mL) in ethanol80% inhibition of biofilm formation* Microorganisms were isolated from consortia and the effects of antimicrobials were assessed on individual cultures.


So, to suppress the growth of polymicrobial biofilms consisting of the yeasts *Rhodotorula mucilaginosa*, *Candida tropicis*, *C. krusei* and *C. kefyr* and bacterial strains *Listeria monocytogenes*, *Salmonella enterica* or *Escherichia coli*, formed on ultrafiltration membranes and stainless steel surfaces on the lines for the production of apple juice, the yeast quorum molecule farnesol was used in combination with natamycin [102]. Farnesol regulates yeast morphogenesis by inhibiting the transition of yeast to the mycelial form. In biofilms, bacteria preferentially associate with hyphae via the fungal adhesin Als3p, which is not expressed in the yeast form.

It is also known that farnesol exhibits selective antibacterial activity against some gram-positive bacterial cells, causing an ionic imbalance in their cytoplasmic membrane and damaging its integrity. In addition, farnesol affects the secretion and activity of beta-lactamases and reduces the amount of free lipid transporter C55 in bacteria, which leads to a subsequent slowdown in the transport of the peptidoglycan monomer precursor across the cell membrane.

It has been shown that when biofilms are treated with farnesol in combination with natamycin, suppression of filamentous forms of yeast, destruction of the matrix and detachment of yeast cells are observed. The action of farnesol was more effective against yeast in the early stages of biofilm formation. It was noted that bacteria were also affected in mixed biofilms, and this effect depended on the type of bacteria. The most sensitive cells were *L. monocytogenes*. In the case of gram-negative bacteria, inhibition by farnesol, even in combination with natamycin, was not so obvious [102].

In another study, when mixed biofilms of *Candida* and *E. coli* cells were exposed to farnesol, an inhibitory effect was observed only at its high concentrations (600 µM) and, on the contrary, at lower doses farnesol stimulated the formation of biofilms [98].

The combination of farnesol with carvacrol demonstrated synergistic inhibitory activity against mixed-species biofilms of amphotricin-B-resistant *C. albicans* cells and *Providencia stuartii* or *S. aureus*. Interestingly, farnesol did not inhibit hyphal formation in *Candida* cells in mixed biofilms. In addition, carvacrol itself at a dose of 2 mg/mL demonstrated 75% biofilm inhibition [103]. The likely reason for this is that the antibiofilm activity of terpenoids is associated with membrane damage, as well as inhibition of oxidative phosphorylation and respiratory chain functions, degradation of cell walls, and decreased adhesion of *C. albicans* to solid surfaces.

Farnesol (300 µM) in combination with oxacillin (2 mg/mL) suppressed the development of a mixed biofilm of *C. albicans*/*S. aureus* by 80% [104]. Microphotographs confirmed the heterogeneity of the biofilm and the absence of *Candida* yeast hyphae.

Thus, combining quorum molecules with antimicrobial agents is an effective approach to inhibit mixed polymicrobial biofilms.

#### 2.6.3. Living Cells of Microorganisms and Their Complex Metabolites in the Regulation of Polymicrobial Consortia

It has been established that cyanobacteria [97], bacteria [99] and fungi [100] can act as biological agents to suppress fungi as part of polymicrobial consortia. As a rule, phototrophic microorganisms and bacteria suppress the growth of fungi by releasing substances that are toxic to them.

Thus, the use of the ascomycete *Clonostachys rosea* as a biological agent to suppress the growth of the zygomycete *Zoophagus* sp., which develops in activated sludge, has proven its effectiveness. *C. rosea* spores germinated into unicellular forms and penetrated into the mycelium of the zygomycete, where they used the cytoplasm as their nutrient medium [100]. Next, mycelium was formed, which wrapped around the hyphae and conidia of other fungi, growing inside them.

To combat fungal pathogens (*Fusarium oxysporum* and *Alternaria* sp.), which lead to banana spoilage, the in vivo use of a consortium consisting of *Penicillium* sp., *Bacillus velezensis* and *B. subtilis* has been proposed. These microorganisms synthesized enzymes (chitinase and β-1,3-glucanase) involved in the degradation of structural components of fungal cell walls. Treatment of bananas with a cell suspension (1.5 × 10^8^ CFU/mL of *B. subtilis* cells) and (1.8 × 10^3^ spore/mL of *Penicillium* sp.) under greenhouse conditions reduced the incidence of plant diseases [88]. The antifungal activity of the enzymes was supplemented in this treatment option by other characteristic metabolites of *B. subtilis* (antibiotic-like substances, iron chelators).

One of the proposed approaches to combating polymicrobial infections is the use of probiotic microorganisms and their metabolites [122]. The main mechanism of action of probiotics is their synthesis of various metabolites that exhibit antimicrobial properties (fatty acids, organic acids, H_2_O_2_, bacteriocins), as well as promoting competition for available nutrients and sites of attachment to various surfaces [123].

The most successfully used probiotics against mixed biofilms are cells of lactic acid bacteria of the genus *Lactobacillus*, since biosurfactants of the cells are able to disrupt the cell membrane structure or change the protein conformations. For example, the supernatant of the medium after culturing *Lactobacillus plantarum* cells was used to treat caries resulting from the action of biofilms consisting of *C. albicans* and *S. mutans* cells [95]. As a result, biofilm inhibition reached 75%, and the number of *S. mutans* and *C. albicans* cells decreased by 91.5% and 43.7%, respectively. The antimicrobial effect of the probiotic supernatant was explained by the presence of AMP (plantaricin), biosurfactants that inhibit the adhesion of pathogens by reducing the hydrophobicity of the substrate surface. Moreover, the expression levels of *S. mutans* genes associated with glucosyltransferase activity and genes specific to the hyphal forms of *C. albicans* (HWP1, ALS1 and ALS3) were reduced in the presence of the probiotic supernatant.

To treat oral candidiasis caused by biofilms of mixed species of yeast of the genus *Candida* (*C. tropicalis*, *C. krusei*, *C. parapsilosis*), supernatants of *Lactobacillus gasseri* and *L. rhamnosus* cells were used in vitro [93]. The metabolic activity of cells in both mono- and mixed biofilms decreased by 50–77.6% after the use of probiotic supernatants (30% *v*/*v*). It was confirmed that the formation of yeast biofilms with mixed *Candida* species on the silicone surface was inhibited by 56.1–67.2%.

When comparing three strains of the genus *Candida*, *C. parapsilosis* cells turned out to be the least susceptible to the action of supernatants of *Lactobacillus* cells. *L. rhamnosus* supernatant inhibited the adhesion of mixed biofilms of *C. albicans*, *C. tropicalis*, *S. salivarius*, *Rothia dentocariosa* and *S. epidermidis* on silicone (91%) and caused a 58.4% decrease in cell viability in biofilms. Under the influence of the supernatant, the transition of yeast to the hyphal form was inhibited [96].

Metabolite extracts isolated from the cell-free supernatant of four strains of the genus *Lactobacillus* were used to inhibit biofilms formed by *C. albicans* and *S. aureus*. As a result, the minimum inhibitory concentration for biofilm was 406 μg/mL. Damage to cell membranes of both pathogens has been shown [94].

In another study, *Streptococcus salivarius* bacteria were used to suppress caries-forming biofilms consisting of *S. mutans* and *C. albicans*, since these cells are capable of secreting proteinaceous antimicrobial substances (bacteriocins) [116]. As a result, a 90% kill rate of *S. mutans* cells was observed, as well as biofilm disruption and damage to the surface of bacterial cells. However, the yeast cells were not affected by *S. salivarius*.

A modified biosurfactant (RLmix_Arg), usually produced by *P. aeruginosa* cells, was obtained, which, in combination with Pluronic F-127 (nonionic detergent), was used against biofilms of *C. albicans*/*S. aureus* in impregnated catheters [101]. The biosurfactant RLmix_Arg included in the gel with Pluronic F-127 showed suppressive activity against both yeast and bacteria, preventing the formation of biofilms in the peripheral venous catheter.

In summarizing information on this area of research, it should be noted that the above-mentioned antimicrobial activity of supernatants obtained after cultivating lactic acid bacteria is extremely interesting for practical application. It manifests itself against mixed polymicrobial biofilms due to the synthesis and accumulation of biologically active compounds by probiotics.

These results open up the possibility for wider use of such media with the obvious ease of their preparation, as well as for more detailed studies aimed at better understanding the compositions (combinations) of accumulating metabolites and the effectiveness of their action in relation to various polymicrobial consortia.

## 3. Viruses in the Regulation of the Functioning of Consortia

### 3.1. Bacteriophages

Bacteriophages can be used among various strategies for controlling mixed polymicrobial biofilms [12]. Bacteriophages, exhibiting high specificity for target cells, synthesize endolysins and depolymerases that destroy bacterial cells, as well as enzymes that are capable of hydrolyzing peptidoglycan. The increased density of bacterial cells inside biofilms significantly enhances the spread of phage infection, which leads to the release of new virions and rapid death of bacterial cells. The effectiveness of using bacteriophages against biofilm depends on its age, morphology, biological and chemical composition. It is known that various modification factors of bacteria can make them insensitive to phages. In this regard, the combined use of bacteriophages with traditionally used antibiotics seems to be a more effective approach (Table 8).
microorganisms-12-01650-t008_Table 8Table 8Bacteriophages against polymicrobial consortia.ConsortiaForm and Site (Reason) of PresenceSuppressive CompoundsEffects*C. albicans*/*P. aeruginosa* [124]Biofilms, cystic fibrosisLytic *Pseudomonas* phage *Motto* (10^9^ PFU (plaque forming units)/mL) combined with cefotaxime, ciprofloxacin, gentamicin, meropenem and tetracycline (128 μg/mL), fuconazole (64 μg/mL)Decrease of *P. aeruginosa* cells in the biofilm without notable influence on the *C. albicans* cells*C. albicans*/*S. aureus* [125]Biofilm, bloodstream infectionsBacteriophages vB_SauM-A (A) and vB_SauM-D (D) (10^7^ PFU/mL) combined with ciprofloxacin (32 μg/mL)82% decrease in *S. aureus* cells in the presence of *C. albicans* in the biofilm 


For example, the effect of a combination of lytic *Pseudomonas* phage *Motto* in combination with various antimicrobial agents (cefotaxime, ciprofloxacin, gentamicin, meropenem and tetracycline) was studied to suppress yeast-bacterial biofilms formed by *C. albicans* and *P. aeruginosa* cells [124]. As a result of the treatment, complete eradication of the biofilm was not confirmed. It turned out that combinations of phage with antibiotics were effective only against mono-biofilms formed by the bacteria *P. aeruginosa*, but their effect on *C. albicans* cells was subtle.

The result of an assessment of the effectiveness of using two bacteriophages vB_SauM-A and vB_SauM-D (10^7^ PFU/mL) in combination with ciprofloxacin (32 mg/L) against biofilms formed by *S. aureus* and *C. albicans* is known [125]. The use of this triple combination made it possible to reduce the number of cells by ≥90%. However, in a mixed biofilm of these bacteria with yeast, the eradication efficiency decreased under the same conditions to 82%. While *S. aureus* cells were eliminated from the biofilm, the number of yeasts decreased slightly due to the action of phage depolymerases, which led to structural changes in the biofilm and the release of yeast cells from the matrix.

In general, given the limited data on the use of bacteriophages against mixed polymicrobial consortia involving fungal cultures, we can conclude that, so far, no effective solutions have been found, even in combination with antibiotics and quorum molecules [124]. In this regard, it may be worth looking for potential good solutions in mycoviruses research.

### 3.2. Mycoviruses as Biological Control Agents

To date, about 200 mycoviruses have been most fully characterized, which are currently classified in 23 families and 1 unclassified genus by the International Committee on Taxonomy of Viruses (https://talk.ictvonline.org/, accessed on 11 May 2024) [125,126,127,128]. The use of next-generation sequencing technologies has accelerated the discovery of new mycoviruses [129]. Mycoviral infections cause various effects, from hypovirulence to hypervirulence, and they can often be hidden (latent infections) [130,131,132].

For the development of new strategies for the biocontrol of microbial consortia, those mycoviruses that, in one way or another, reduce the viability and resistance of fungi are of interest [133]. Mycoviral infections lead to changes in the host transcriptome profile through protein-protein interactions and triggering the suppression of antiviral RNA in fungal cells [129]. Mycoviruses and mycovirus-mediated hypovirulence have been mainly studied in phytopathogenic ascomycetes and basidiomycetes. The most famous are *Pezizomycotina* cells, representing the most diverse group of ascomycetes by species, as well as common mold fungi such as *Aspergillus*, and many phytopathogenic and endophytic fungi [134].

The first was a mycovirus of the Hypoviridae family (CHV1), which began to be considered as a means of biological controlling of the ascomycete fungus that causes chestnut blight (*Cryphonectria parasitica*) [135]. The virus belongs to the *Hypoviridae* family, located in the cytoplasm membrane vesicles of their fungal hosts, without a coat protein, and with double-stranded RNA (dsRNA) replication form. DNA methyltransferase (DNMTase) genes play a significant role in the response to hypoviral infection CHV1 [135,136]. Further research in this direction led to the discovery of new hypovirulent strains in other representatives of fungi (Table 9 [135,137,138,139,140,141,142,143,144,145], Figure 7).
microorganisms-12-01650-t009_Table 9Table 9Examples of mycoviruses that cause hypovirulence in the phytopathogen host and can be used as biological control agents.Target Fungi for Mycovirus ActionNegative Effect of the Fungal CellsMycovirus [Reference]Mycovirus Family*Cryphonectria parasitica*Causative agent of chestnut blightCHV-1 *Cryphonectria hypovirus* 1 [135]*Hypoviridae* dsRNA*Sclerotinia** sclerotiorum*Infecting over 400 plant species found worldwide—*Sclerotinia* stem rot or white moldSsHADV-1 *Sclerotinia sclerotiorum* negative-stranded RNA virus 1 [137]*Sclerotiniaceae *ssDNA*Magnaporthe oryzae (Pyricularia oryzae)*Rice blastMoCV1-A *Magnaporthe oryzae* chrysovirus 1-A [138]*Chrysoviridae* dsRNA*Botrytis cinerea* (*Botryotinia fuckeliana*) Grey mould with a necrotrophic lifestyle attacking over 200 crop hostsBcMV10 *Botrytis cinerea* mitovirus 10 [139]*Mitoviridae* + ssRNA*Fusarium oxysporum**Fusarium* wiltFodV1 *Fusarium oxysporum f.* sp. dianthi virus 1 [140]*Chrysoviridae*dsRNA*Botryosphaeria dothidea*Causing the canker and dieback of fruit trees: pear, poplar, apple, walnut, and jujube treesBdCV1 *Botryosphaeria dothidea* chrysovirus 1 [141]*Chrysoviridae* dsRNABdRV1 *Botryosphaeria dothidea* RNA virus 1 [142]*Polymycoviridae*dsRNABdPV1 *Botryosphaeria dothidea* partitivirus 1 [141,143]*Partitiviridae*dsRNABdPV2 *Botryosphaeria dothidea* partitivirus 2 [144]*Partitiviridae* dsRNA*Rosellinia necatrix*White root rot disease (apple, avocado, pears)RnPV10 *Rosellinia necatrix* partitivirus 10 and RnVLV *Rosellinia necatrix* virga-like virus [145]*Partitiviridae* dsRNAVirgaviridae-like+ssRNA


The use of *Fusarium oxysporum f.* sp. dianthi virus 1 (FodV1) can be successful in the biological control of *Fusarium* infections caused by the fungus *F. oxysporum* [140]. The phytotoxicity of representatives of *Fusarium oxysporium* is known both for an individual culture and as part of a consortium [146].

More than 10 mycoviruses have been discovered that infect *Botryosphaeria dothidea*, which is a phytopathogen causing ring rot of pears and other fruit trees. Some of them, for example, *Botryosphaeria dothidea* chrysovirus 1 (BdCV1), *Botryosphaeria dothidea* RNAvirus 1 (BdRV1), *Bipolaris maydis* botybirnavirus 1 BdEW220 (BmRV1-BdEW220), *Botryosphaeria dothidea* partitivirus 1 (BdPV1) and *Botryosphaeria dothidea* partitivirus 2, reduce the virulence of the fungus *Botryosphaeria dothidea* [141,142,143,144,147]. It is known that *B. dothidea* infects fruits together with other fungal pathogens, which means that it is a potential participant of microbial consortia. Thus, two fungal strains of *B. dothidea* in the presence of *Colletotrichum gloeosporioides* were identified as the main pathogens isolated from rotten apples stored without packaging. Essential oils of cinnamon and clove showed high inhibitory activity against the growth of mycelium of this consortium in vitro [148]. Vapors of essential oils at a concentration of 60 μL/L significantly reduced the frequency and diameter of fungal lesions in vivo [148]. In this regard, we should expect high effectiveness from combining mycoviruses and essential oils to regulate the growth of polymicrobial consortia that include *B. dothidea*.

Various combined options for using mycoviruses with other biological objects can be considered to enhance the effect of controlling the growth of fungal consortia. In particular, the use of *Magnaporthe oryzae* chrysovirus 1-A (MoCV1-A) [138] in combination with biological agents in the form of bacteria (*Streptomyces* sp. [149] or *Bacillus* sp. [150]) which, like mycovirus, are able to suppress the growth of this fungus, can be considered in order to suppress the activity of the fungus *Magnaporthe oryzae*, which infects rice plantations.

To control the growth of *Botrytis cinerea* (*Botryotinia fuckeliana*), which secretes enzymes and toxins that cause the death of plant cells [151], the use of *Botrytis cinerea* mitovirus 10 (BcMV10) [135] in combination with bacterial (*Pseudomonas fluorescens*, *Bacillus thuringiensis* UM96, *Streptomyces* spp.) or fungal (*Trichoderma harzianum*, *T. viride*, *T. virens*, *Ulocladium* spp., *Clonostachys rosea*, *Gliocladium catenulatum*, *Saccharomyces cerevisiae*, *Wickerhamomyces anomalus*, *Metschnikowia pulcherrima*, *Aureobasidium pullulans*) cells can be considered [152].

*Sclerotinia sclerotiorum* negative-stranded RNA virus 1 (SsNSRV-1) is the first mycovirus with a negative sense ssRNA genome [137,153]. Virions of this virus, associated with the hypovirulence of the fungus *S. sclerotiorum*, can directly infect its hyphae, which lead to hypovirulence of the fungal host [137]. *S. sclerotiorum* is a pathogenic fungus that causes white mold (cotton rot) on more than 600 plants, including many of the most important crops (canola, soybean, sunflower, and others).

When studying combined approaches to control the growth of *S. sclerotiorum* using the mycovirus SsNSRV-1, the greatest effect should be expected in the presence of *Trichoderma harzianum* and *T. asperellum* cells. In the presence of a consortium of fungi *Trichoderma* sp., a significant increase in the activity (level of synthesis) of eggplant protective enzymes (phenylalanine ammonia lyase, peroxidase and polyphenol oxidase) and inhibition of the growth of the pathogen *S. sclerotiorum* were noted [154].

When creating a combination of drugs involving mycoviruses, it should sometimes count on the possible effect of not just one virus but a combination of mycoviruses. Thus, it was found that *Rosellinia necatrix* partitivirus 10 (RnPV10) and *Rosellinia necatrix* virga-like virus (RnVLV) have a hypovirulent effect on the fungus *Rosellinia necatrix*, causing root rot in avocado, only in a joint combination [145]. At the same time, an enhanced effect from the biological suppression of the growth of *R. necatrix* should be expected with the combined use of mycoviruses with bacteria of the genus *Bacillus* containing genes necessary for the biosynthesis of lipopeptides, polyketides and bacilisin, for which the effect of suppressing the growth of this fungal pathogen has already been established [155].

Of interest are not only mycoviruses that directly infect pathogenic fungi but also viruses that inhibit the synthesis of phytotoxins by fungi, as well as mycoviruses that encode killer toxins and suppress the development of endophytic fungi with negative effects on herbivores [127,156].

*Stemphylium lycopersici* alternavirus 1 (SlAV1) was isolated from a necrotrophic plant pathogen *S. lycopersici* infecting tomatoes. SlAV1 (family Alternaviridae) significantly reduced the activity of fungal polyketide synthase, which catalyzes the biosynthesis of altersolanol A (a polyketide phytotoxin). If fungi do not accumulate altersolanol A, then they lose pathogenicity towards tomatoes and lettuce [156] since altersolanol A is a kinase inhibitor, which induces apoptosis of cells.

Some strains of yeast produce protein toxins that are lethal to sensitive strains of fungi. Yeasts that secrete such toxins are called “killer-yeast”, and the proteins they secreted are called “killer-toxins” [157].

To date, more than 100 species of killer-yeasts have been described [158]. Toxins-producing killer strains were found among *S. cerevisiae*, *Hanseniaspora uvarum*, *Ustilago maydis* and *Zygosaccharomyces bailii*. The majority of yeast killer-toxins are encoded in the nucleus. Some of them are encoded by selfish genetic elements consisting of viral or virus-like dsDNA (in *Kluyveromyces lactis* and *Pichia acacia*) or dsRNA molecules. *Zygosaccharomyces bailii* virus Z (dsRNA family *Amalgaviridae*) of yeast *Z. bailii* encoding a preprozygocin precursor is processed to the biologically active toxin (zygocin) on sensitive fungal cells. Zygocin induces apoptotic yeast cell response at relatively low concentrations while it determines necrotic, toxin-specific cell killing independent from yeast caspase 1 [159]. The K1, K2, and K28 toxins are usually encoded by several cytoplasmically genetic satellite dsRNAs (M1, M2, and M28), which are encapsulated with virus-like particles [160].

Endophytic fungi are in symbiotic association with various cereal plants without causing any external signs of infection. Plants provide mushrooms with food and a stable habitat. A number of mycoviruses are capable of converting pathogenic fungi into beneficial endophytes [161]. At the same time, it is known that some endophytic fungi secrete alkaloids that protect the plant both from insect pests and, partially, from herbivorous mammals and increase the resistance of the host plant to diseases and adverse environmental factors. Animals that are fed grass with a high proportion of endophyte contamination develop symptoms of poisoning, including death. Therefore, it is important to control the growth and development of such endophytic fungi in agriculture. One of the options for biocontrol can be considered a viral infection of endophytes, causing inhibition of the development of these fungi.

Thus, the potential areas for the use of mycoviruses as agents of biocontrol and suppression of the growth of pathogenic fungi, including as part of consortia, are quite diverse but are limited mainly by the routes of transmission of viruses. Mycoviruses can only spread intracellularly through hyphal anastomosis (horizontal transmission) or sporulation (vertical transmission) [127]. So far, there is little research in this direction, but they are undoubtedly relevant and promising.

## 4. Analysis of Wide-Spread Polymicrobial Consortia with Fungi and the Approaches to Their Suppression and Elimination

Analyzing the problems and means used to suppress the activity of various polymicrobial consortia, it was interesting to identify among them those options that cause the greatest concern from their presence and attract the maximum attention of researchers. At the same time, it was interesting to identify consortia by their composition and by the areas of formation of these consortia (Figure 8 and Figure 9).

It appeared that food-oriented consortia form a distinguishable cluster (in green color, Figure 8). At the same time, susceptible consortia for medicine are divided into two clusters: centered on *C. albicans* and *C. tropicalis*, respectively. Widely studied *S. aureus*, *P. aeruginosa* and *E. coli* strongly gravitate to *C. albicans*. It should be noted here that *S. aureus* and *E. coli* can form natural stable consortia with *C. albicans* [28,31,33,37,42,45,54,56,57,65,66,70,74,75,76,85,86,94,98,101,104,125]. However, combinations of *P. aeruginosa* with *C. albicans* are unstable [43,80,124] and their artificial consortia, as a rule, deteriorate quite quickly with the domination of *P. aeruginosa*.

Surprisingly, *S. aureus* and *E. coli* hold central places while connecting medically oriented investigations to industrial ones (particularly the food industry [87,92]).

Filamentous fungi of susceptible microbial consortia are allocated in the dispersed halo around central parts with the single-entry point of *A. fumigatus* group (in orange color) connected to *P. aeruginosa* [48] or *Stenotrophomonas maltophilia* [67]. Yet two single strains (*S. apiospermum* and *F. oxysporum*) are connected to *P. aeruginosa* and *C. albicans* [29,49], while *F. solani* is connected to both *Staphylococcus* [52].

Interestingly, almost all mixed consortia of susceptible filamentous fungi (Figure 8) didn’t contain other species [40,59,61,69,88,89,90]. Moreover, the greatest number of resistant consortia (Figure 9) were also formed by filamentous fungi only [61,69]. Unfortunately, the total number of such resistance reports is tiny and is likely to be predetermined by observational exception (i.e., such results are not included in the articles).

An analysis of the data from all Table 1, Table 2, Table 3, Table 4, Table 5, Table 6, Table 7 and Table 8 presented in this review was carried out, reflecting various options for suppressing the growth and metabolic activity of cells in mixed consortia, which include filamentous fungi or yeast, up to their destruction. This made it possible to estimate the percentage distribution of modern developments according to the different approaches used to solve the problem in different processes (Figure 10) and draw several main conclusions about the existing preferences in the use of certain means:-Today, AMPs are studied exclusively for medical purposes to suppress the discussed consortia with pathogens.-Physical and chemical methods are being studied for use in regulating water purification processes, as well as in medicine.-Agriculture turned out to be focused on studying the negative impact of metal NPs on the vital activity of cells in mixed consortia.-Microbial cells and enzymes are traditionally of interest for suppressing spoilage processes in the food industry.-All of the approaches to suppressing microbial consortia covered in the review have been tested in medicine, including many of them used to preserve cultural heritage sites, prevent their degradation, and for environmental purposes (to free water purification systems from biofilms).-Metal NPs and physicochemical methods are not popular for suppressing biofilms for medical purposes, probably due to the fact that NPs are toxic to any cells, and physicochemical methods are characterized by low efficiency; the use of such means and methods is considered for sanitary purposes.-Combinations of different antimicrobial agents are widely studied and have a greater suppressive effect in various fields of application against consortia of different compositions in comparison with individual compounds. However, combinations, as well as physicochemical methods of suppression, have been studied very little in the interests of agriculture and combating biocorrosion. Obviously, the potential for using possible combinations has not yet been revealed, and there is a wide field for scientific activity and experimental research.

When discussing the limitations for obtaining a positive effect from the combined use of antibiotics, the provoked appearance of microorganisms (including fungi) with multidrug resistance should be noted. There are several main reasons for its appearance:(1)This is due to the use of several antibiotics in reduced concentrations to overcome their total toxicity problem [162].(2)A combination of antibiotics is used when one of them is already ineffective, and the start of treatment with a combination of antibiotics is delayed due to the presence of pathogens in high concentrations and the QS state. For example, a combination of several antibiotics (cefazolin, gentamicin and vancomycin) suppressed biofilm formation on an orthopedic implant during the first 3 days after infection but did not have a positive effect if the same combination was used after 7 days of infection development [163].(3)There are known risks of the combined use of two or more antibiotics due to the occurrence of undesirable interference between them and the manifestation of drug-drug antagonism [164]. Thus, during clinical trials, it was shown that combinations consisting of antibiotics of group I (penicillin, streptomycin, bacitracin, neomycin and polymixin) and group II (chloramphenicol, tetracyclins and erythromycin) are often antagonists. Observations of individual cells confirmed the antagonism between bacteriostatic and bactericidal antibiotics [165]. At the same time, it is also known that the effectiveness of a combination of antibiotics can be antagonistic in one concentration range but synergistic in another, however, this should be established experimentally [166].(4)Often, the use of combinations of antibiotics leads to their appearance in the environment since the degradation efficiency of combined antibiotic variants is quite low in wastewater treatment plants [7].

Thus, there are restrictions for the use of several antibiotics for combined therapy; therefore, the task of combining enzymes with antibiotics instead of antibiotics with each other arises.

To date, extensive information has been accumulated, which indicates that fungi can cause commensurate or even greater harm to humanity or areas closely related to human activity than bacteria. The reasons here are as follows:-The results of the analysis of complications in a significant number of patients after the COVID-19 pandemic indicate that fungi of the genera *Candida*, *Aspergillus*, *Mucor* cause opportunistic infections in patients with weakened immunity [167,168]. At the same time, fungi are often associated with bacteria and show high resistance to antimicrobial drugs;-In patients with Parkinson’s disease, immunohistochemistry and specific antibodies allowed the revealing of both bacteria (genera *Streptococcus* and *Pseudomonas*) and fungi (genera *Botrytis*, *Candida*, *Fusarium* and *Malassezia*) in brain tissues [169] that initiates the need to correct treatment strategies for such patients;-In dental laboratories, it was found that even after disinfection of pumice, there remains a high probability of cross-infection for technicians, dentists and patients with both bacteria (*Acinetobacter lowffi*, *Bacillus cereus*, *Staphylococcus aureus*, *Pseudomonas aeruginosa*, *Diphteroids*, *Serratia mercescens*, *Enterobacter aerogenes*, *Morganella morgani*, *Providencia rettgeri*, *Staphy-lococcus albidus* and *Streptococcus sanguis*), and fungi (*Candida* sp., *Aspergillus niger*, *Fusarium* sp., *Aspergillus flavous*, *Cephalosporium* sp. and *Pencillium* sp.) [170,171];-Soil bacterial (*Streptomyces* spp.) and fungal cells (*Trichoderma* spp.) can cause human skin diseases and a number of other dangerous diseases [172];-Fungi of the genera *Aspergillus*, *Alternaria*, *Penicillium*, *Aureobasidium* are capable of causing bio-damage to construction sites [173,174]. Fungal spores and metabolites (mycotoxins) that cause the development of upper and lower respiratory tract infections, allergic reactions and poisoning can cause bronchial irritation and allergy, broncho pulmonary mycoses, and hypersensitivity pneumonitis [175]. In chronic rhinosinusitis, fungal and bacterial biofilms are usually found, which can cause a chronic and antimicrobial-resistant stage of the disease [176]; the presence of mycotoxins in food and feed is dangerous due to their toxicity and carcinogenesis development [109,111];-Bacteria, as the main microorganisms present in the human body, are studied more widely, whereas databases of complete genomes of fungi present in the human body are less well provided [177]; this lack of information underlies the development of resistant dysbiosis;-In the process of observed climatic changes, viral and bacterial diseases as a potential cause of epidemics and pandemics may fade into the background and give way to fungi since it is fungi that can pose an equal or even greater threat: there are no vaccines against fungal pathogens yet, the arsenal of antifungal drugs is extremely limited, including even AMPs [109,178,179], and fungi can live saprotrophic, producing a large number infectious spores, without requiring direct contact with the affected object, have a unique ability to adapt to new conditions, including temperature conditions [180]. Taking all this into account, fungal pathogens have recently been included for the first time in the “World Health Organization (WHO) fungal priority pathogens list”, compiled from 19 groups of fungal microorganisms associated with a serious risk of human mortality or morbidity [181].

Thus, when studying the possibilities of preserving the sustainable development of humanity, fungi deserve no less attention than bacteria, especially as part of mixed consortia.

## 5. Conclusions

Polymicrobial consortia, especially those with fungal cells, pose a serious challenge in terms of managing such complex biosystems. Cells in such consortia may, basically, not always exhibit predictable behavior and, even more so, a reaction to the applied suppressive effects on a microbial community of complex composition. Cells may exhibit greater resistance to currently used antimicrobials in such consortia compared to planktonic cells or monogenic biofilms. Although the development and search for new antimicrobial agents of various chemical natures characterized by low toxicity are currently underway, it is obvious that the most effective solution seems to be the combined use of already known and well-characterized compounds that suppress the growth and metabolic activity of cells. It is also advisable to combine different methods and approaches to cell suppression. Not only the destruction of biofilms is important but also the death of cells inside them, since the death of only one type of cell in a consortium can lead to an increase in the number of other cells and the formation of new biofilms. Despite the fact that various effective solutions have been shown in vitro (for example, the use of metal NPs and antimicrobial agents of plant origin, cell metabolites in combination with antimicrobial agents acting on several targets simultaneously), studies of such combinations in vivo are important. Today, it is still impossible to clearly predict in advance the positive synergistic effect of combining different compounds when acting on polymicrobial consortia, so experiments are necessary. Nevertheless, the very possibility of combining individual compounds can be assessed theoretically. For this, as well as for the development of synthetic antimicrobial compounds and biomimetics, modern methods of computer molecular modeling should be used.

## Figures and Tables

**Figure 1 microorganisms-12-01650-f001:**
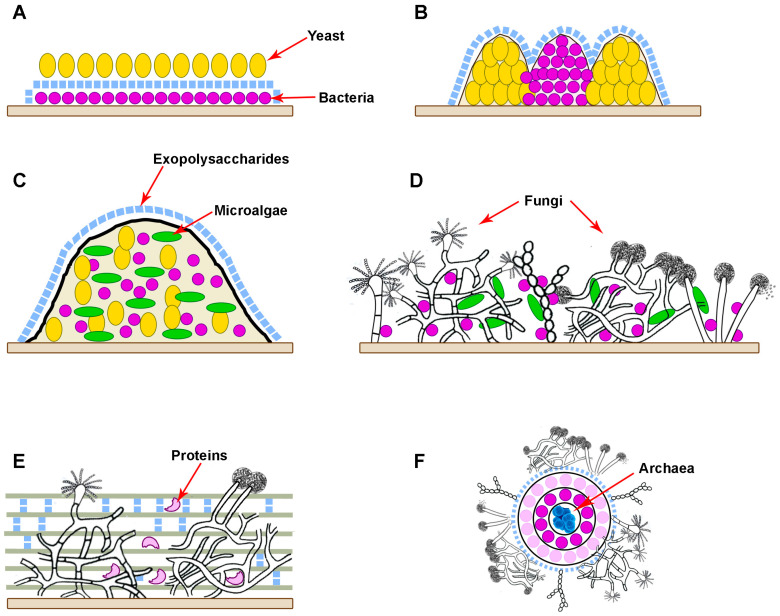
Scheme of different types of polymicrobial biofilms containing various fungi: microorganisms can coexist having a layer-by-layer arrangement (**A**), they can be mixed partially (**B**) or deeply (**C**) when one type of cell is located in the upper layer, and another type in the lower (**D**); the fungal spores can appear in the biofilm matrix (**E**), or fungi can cover the layers formed by other cells (**F**).

**Figure 2 microorganisms-12-01650-f002:**
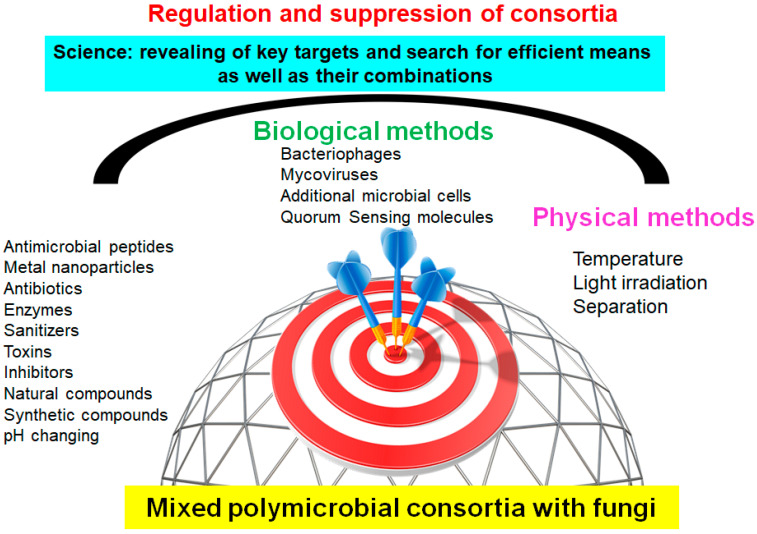
Basic methods and means of suppressive effects on polymicrobial consortia containing filamentous fungi or yeasts.

**Figure 3 microorganisms-12-01650-f003:**
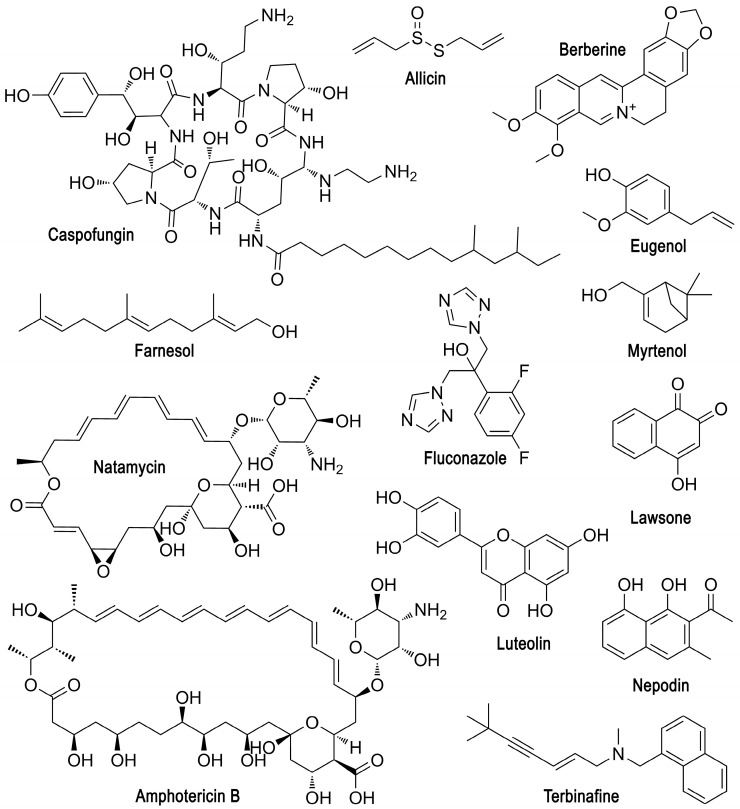
Chemical structures of some low-weight organic effectors investigated for suppression of mixed consortia.

**Figure 4 microorganisms-12-01650-f004:**
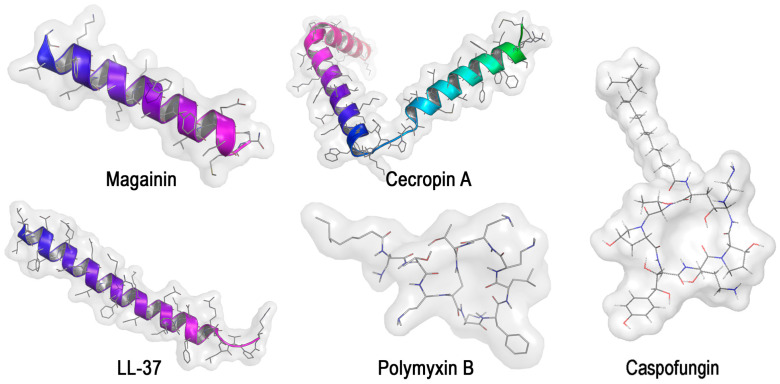
Representation of some AMPs mentioned in Table 2. Structures were obtained from PDB (4MGP and 5NNM) and Uniprot (P01507) and visualized using PyMol (ver. 1.7.6, Schrödinger, LLC, New York, NY, USA).

**Figure 5 microorganisms-12-01650-f005:**
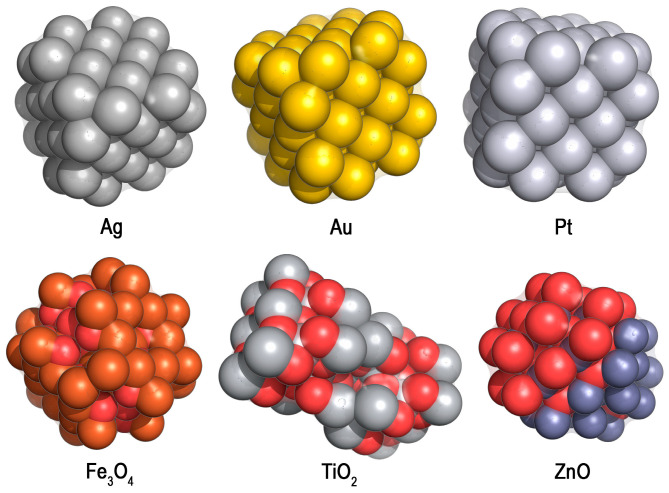
Structures of some metal-containing NPs mentioned in Table 4. The corresponding crystal structures were obtained from Cambridge Crystallographic Data Centre (CCDC 1741252, 1744485, 1763757, 1612598, 1481933 and 13950), expanded to around 1.2–1.5 nm in Mercury (v. 4.2.0, CCDC, Cambridge, UK) and visualized in PyMol. In this regard, many researchers propose their use to suppress the metabolism of pathogens in biofilms, but it has been found that consortia formed in the presence of heavy metals are tolerant to their negative effects [7].

**Figure 6 microorganisms-12-01650-f006:**
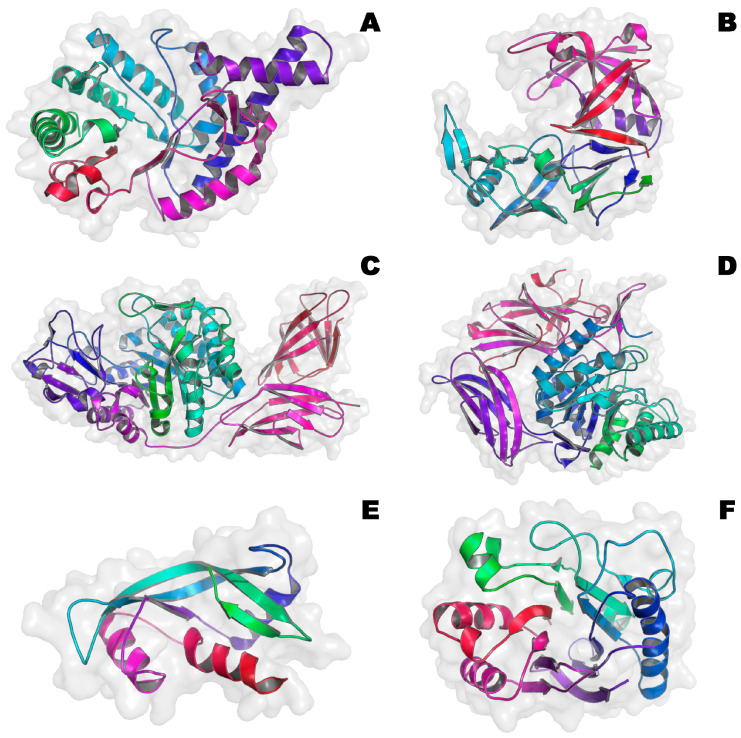
Structures of some enzymes mentioned in Table 6: hyaluronoglucosaminidase (PDB 1FCV) (**A**), protease (PDB 3PVK) (**B**), chitinase (PDB 6BT9) (**C**), β-glucuronidase (PDB 5C70) (**D**), RNAse (PDB 5ARK) (**E**), DNAse (PDB 1DNK) (**F**).

**Figure 7 microorganisms-12-01650-f007:**
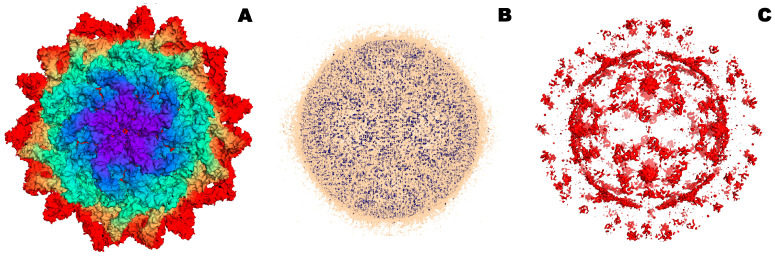
Structures of some representative viruses mentioned in Table 9: *Penicillium stoloniferum* partitivirus F (PDB 3ES5) (**A**), *Cryphonectria nitschkei* chrysovirus 1 (EMD-2062) (**B**), *Fusarium poae* virus 1 (EMD-5171) (**C**).

**Figure 8 microorganisms-12-01650-f008:**
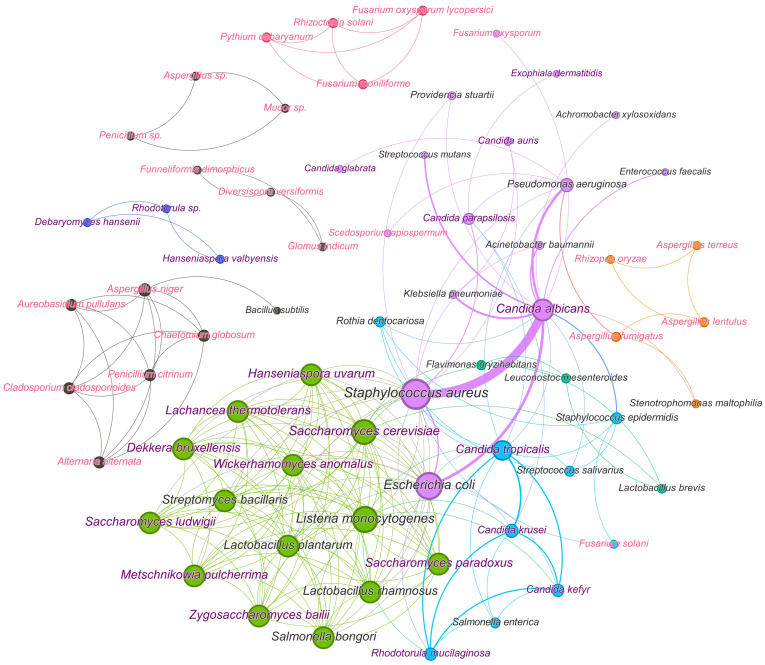
Network analysis of microbial consortia susceptible to antimicrobial treatment. The lines connect the microorganisms from different consortia discussed in the Table 1, Table 2, Table 3, Table 4, Table 5, Table 6 and Table 7. Names of bacterial, fungal and yeast strains are labeled by black, pink and magenta color, respectively. The size of nodes and their labels is proportional to the ranking degree; the thickness of edges is proportional to their weight. Nodes belonging to different modularity classes are colored in groups accordingly. The analysis was realized in Gephi (v.0.10.1, available at https://gephi.org/users/download/, accessed on 12 May 2024).

**Figure 9 microorganisms-12-01650-f009:**
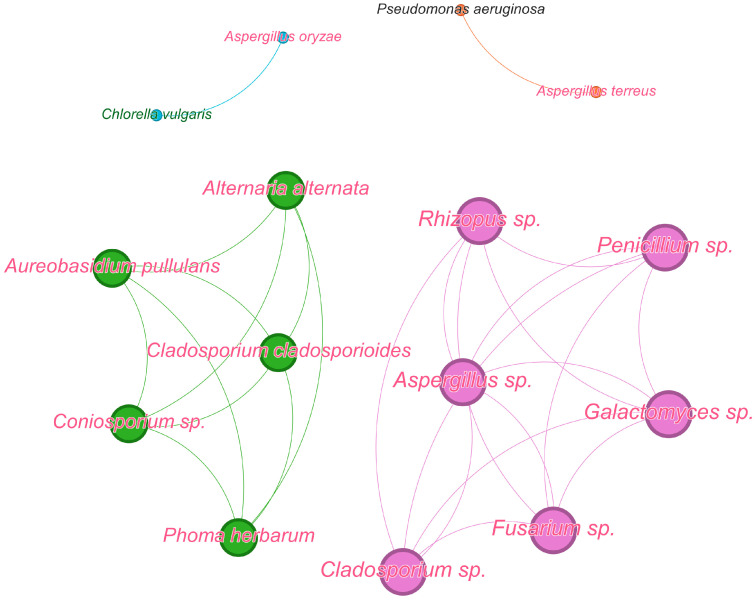
Network analysis of microbial consortia resistant to antimicrobial treatment. Names of bacterial, fungal and microalgal strains are labeled by black, pink and green color, respectively. The size of nodes and their labels is proportional to the ranking degree; the thickness of edges is proportional to their weight. Nodes belonging to different modularity classes are colored in groups accordingly. The analysis was realized in Gephi (v.0.10.1, accessible at https://gephi.org/users/download/, accessed on 12 May 2024).

**Figure 10 microorganisms-12-01650-f010:**
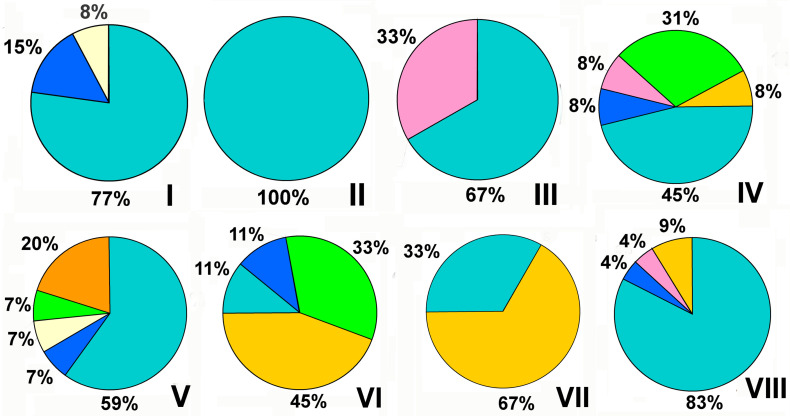
Percentage distribution of modern developments based on various approaches (I—non-peptide antimicrobial compounds of plant and animal origin, II—AMPs, III—enzymes, IV—microorganisms and their metabolites, V—various antimicrobial agents and disinfectants, VI—heavy metal ions and metal nanoparticles, VII—physical and chemical methods of influence, VIII—combined methods for inhibition of polymicrobial consortia) by areas of proposed application: ■—medicine, ■—preservation of cultural heritage sites, ■—fight against biocorrosion, ■—food industry, ■—agriculture, ■—water purification and soil bioremediation. The figure was constructed using the data presented in Table 1, Table 2, Table 3, Table 4, Table 5, Table 6 and Table 7.

## Data Availability

Not applicable.

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
