# Peer review of "“Stop, Little Pot” as the Motto of Suppressive Management of Various Microbial Consortia"

_microorganisms, 2024, doi:10.3390/microorganisms12081650_

Round 1
Reviewer 1 Report
Comments and Suggestions for Authors
Papers discussing microbial consortia are always intriguing. This topic represents one of the most mysterious aspects surrounding humanity, with much more unknown than known. Furthermore, microbial consortia are anticipated to provide excellent sources for addressing some of the most significant challenges faced by humanity, such as antibiotic resistance and biofilm formation. The paper is meticulously and comprehensively written, demonstrating excellent quality and readability. It provides sufficient information required for a review paper. However, I would like to offer a few suggestions to further enhance the paper.
1. The explanation of biofilms in the Introduction section is spread across several paragraphs and contains some redundant information. This should be avoided. I recommend consolidating the paragraphs related to biofilms.
2. Please describe the limitations of antibiotic combination therapy. Without this, there is a risk of misunderstanding combination therapy as a universal solution.
3. What is the basis for the view that fungi pose as much harm to humanity or are as closely related to humans as bacteria? Please elaborate on this point in the relevant section.
4. The paper covers a broad range of topics, including antimicrobial peptides (AMPs). Please specify and emphasize which pathogenic organisms are currently within the control range of AMPs.
Author Response
On behalf of the authors, I would like to express my great gratitude to the Reviewer for the interest to our article, kind attitude towards the authors, a good assessment of our work and valuable suggestions for improving the quality of the review.
Comments and Suggestions for Authors from Reviewer #1
Papers discussing microbial consortia are always intriguing. This topic represents one of the most mysterious aspects surrounding humanity, with much more unknown than known. Furthermore, microbial consortia are anticipated to provide excellent sources for addressing some of the most significant challenges faced by humanity, such as antibiotic resistance and biofilm formation. The paper is meticulously and comprehensively written, demonstrating excellent quality and readability. It provides sufficient information required for a review paper. However, I would like to offer a few suggestions to further enhance the paper.
- The explanation of biofilms in the Introduction section is spread across several paragraphs and contains some redundant information. This should be avoided. I recommend consolidating the paragraphs related to biofilms.
Response. Based on the recommendation of the Reviewer, part of the information on the diversity of biofilms formed by microorganisms with the participation of mycelial fungi and yeast was given in the form of a new figure 1, and the text was partially shortened in the Introduction. In this regard, the numbering of the figures in the text has changed.
- Please describe the limitations of antibiotic combination therapy. Without this, there is a risk of misunderstanding combination therapy as a universal solution.
Response. Discussing the limitations for obtaining a positive effect from the combined use of antibiotics, it should be noted the provoked appearance of microorganisms (including fungi) with multidrug resistance. There are several main reasons for its appearance:
1) this is due to the use of several antibiotics in reduced concentrations to overcome the problem of their total toxicity [Kim, S.; Lieberman, T.D.; Kishony, R. Alternating antibiotic treatments constrain evolutionary paths to multidrug resistance. PNAS Proc. 2014, 111(40), 14494-14499. https://doi.org/10.1073/pnas.1409800111];
2) a combination of antibiotics is used when one of them is already ineffective, and the start of treatment with a combination of antibiotics is delayed due to the presence of pathogens in high concentrations and the QS condition. For example, a combination of several antibiotics (cefazolin, gentamicin and vancomycin) suppressed biofilm formation on an orthopedic implant during the first 3 days after infection, but did not have a positive effect after 7 days of infection development [Tomizawa, T.; Nishitani, K.; Ito, H.; Okae, Y.; Morita, Y.; Doi, K.; Saito, M.; Ishie S.; Yoshida S.; Murata K.; Yoshitomi H.; Kuroda Y.; Matsuda, S. The limitations of mono‐and combination antibiotic therapies on immature biofilms in a murine model of implant‐associated osteomyelitis. JOR, 2021, 39(2), 449-457. https://doi.org/10.1002/jor.24956];
3) известны риски комбинированного применения двух и более антибиотиков из-за возникновения нежелательной интерференции между ними and drug-drug antagonism manifestation [Coates, A.R.; Hu, Y.; Holt, J.; Yeh, P. Antibiotic combination therapy against resistant bacterial infections: synergy, rejuvenation and resistance reduction. Expert Rev. Anti-Infect. Ther., 2020, 18(1), 5-15. https://doi.org/10.1080/14787210.2020.1705155]. Thus, during clinical trials, it was shown that combinations consisting of antibiotics of group I (penicillin, streptomycin, bacitracin, neomycin and polymixin) and group II (chloramphenicol, the tetracyclins and erythromycin) are often antagonists. Observations of individual cells confirmed the antagonism between bacteriostatic and bactericidal antibiotics [Ocampo, P.S.; Lázár, V.; Papp, B., Arnoldini, M., Abel zur Wiesch, P., Busa-Fekete, R., Fekete G., Pál C., Ackermann M., Bonhoeffer, S. Antagonism between bacteriostatic and bactericidal antibiotics is prevalent. AAC, 2014, 58(8), 4573-4582. https://doi.org/10.1128/aac.02463-14]. At the same time, it is also known that the effectiveness of a combination of antibiotics can be antagonistic in one concentration range, but synergistic in another, but this is being established experimentally [Singh, N.; Yeh, P.J. Suppressive drug combinations and their potential to combat antibiotic resistance. J. Antibiot. 2017, 70(11), 1033-1042. https://doi.org/10.1038/ja.2017.102];
4) often, the use of combinations of antibiotics leads to their appearance in the environment, since the degradation efficiency of combined antibiotic variants is quite low in wastewater treatment plants [Efremenko, E.; Stepanov, N.; Senko, O.; Maslova, O.; Lyagin, I.; Aslanli, A. Progressive biocatalysts for the treatment of aqueous systems containing pharmaceutical pollutants. Life 2023, 13, 841. https://doi.org/10.3390/life13030841].
Thus, restrictions for the use of combined therapy used several antibiotics, and therefore, the tasks arise of combining not antibiotics with each other, namely enzymes with antibiotics.
This text with references was added to the article.
- What is the basis for the view that fungi pose as much harm to humanity or are as closely related to humans as bacteria? Please elaborate on this point in the relevant section.
Taking in account the question of the Reviewer, the following text with corresponding references was added to the article:
“To date, extensive information has been accumulated, which indicates that fungi can cause commensurate or even greater harm to humanity or areas closely related to human activity than bacteria. The reasons here are as follows:
- the results of the analysis of complications in a significant number of patients after the COVID-19 pandemic indicate that fungi of the genera Candida, Aspergillus, Mucor cause opportunistic infections in patients with weakened immunity [Kubin, C.J.; McConville, T.H.; Dietz, D.; Zucker, J.; May, M.; Nelson, B.; Istorico E.; Bartram L.; Small-Saunders, J.; Magdalena, E.; Sobieszczyk, M.E.; Gomez-Simmonds, A.; Uhlemann, A.C. Characterization of bacterial and fungal infections in hospitalized patients with coronavirus disease 2019 and factors associated with health care-associated infections. Open Forum Infect. Dis., 2021, 8(6), ofab201. https://doi.org/10.1093/ofid/ofab201; Meawed, T.E.; Ahmed, S.M.; Mowafy, S.M.; Samir, G.M.; Anis, R.H. Bacterial and fungal ventilator associated pneumonia in critically ill COVID-19 patients during the second wave. J. Infect. Public. Health 2021, 14(10), 1375-1380. https://doi.org/10.1016/j.jiph.2021.08.003]. At the same time, fungi are often associated with bacteria and show high resistance to antimicrobial drugs;
- in patients with Parkinson's disease, immunohistochemistry and specific antibodies allowed revealing of both bacteria (genera Streptococcus and Pseudomonas) and fungi (genera Botrytis, Candida, Fusarium and Malassezia) in brain tissues [Pisa, D.; Alonso, R.; Carrasco, L. Parkinson's disease: a comprehensive analysis of fungi and bacteria in brain tissue. Int. J. Biol. Sci., 2020, 16(7), 1135. https://doi.org/10.7150/ijbs.42257]; that initiates the need to correct treatment strategies for such patients;
- in dental laboratories, it was found that even after disinfection of pumice, there remains a high probability of cross-infection for technicians, dentists and patients with both bacteria (Acinetobacter lowffi, Bacillus cereus, Staphylococcus aureus, Pseudomonas aeruginosa, Diphteroids, Serratia mercescens, Enterobacter aerogenes, Morganella morgani, Providencia rettgeri, Staphy-lococcus albidus and Streptococcus sanguis), and fungi (Candida sp., Aspergillus niger, Fusarium sp., Aspergillus flavous, Cephalosporium sp. and Pencillium sp.) [Vojdani, M.; Zibaei, M. Frequency of bacteria and fungi isolated from pumice in dental laboratories. J Res Health Sci, 2023, 6(2), 33-38.; Firoozeh, F.; Zibaei, M.; Zendedel, A.; Rashidipour, H.; Kamran, A. Microbial contamination of pumice used in dental laboratories. Healthcare in Low-resource Settings. 2013, 1(1), 5. https://doi.org/10.4081/hls.2013.e5.];
- soil bacterial (Streptomyces spp.) and fungal cells (Trichoderma spp.) can cause human skin diseases and a number of other dangerous diseases [Brevik, E.C.; Slaughter, L.; Singh, B.R.; Steffan, J.J.; Collier, D.; Barnhart, P.; Pereira, P. Soil and human health: current status and future needs. Air, Soil and Water Research, 2020, 13, 1178622120934441. https://doi.org/10.1177/1178622120934441];
- fungi of the genera Aspergillus, Alternaria, Penicillium, Aureobasidium are capable of causing bio-damage to construction sites and artifacts [Di Carlo, E.; Chisesi, R.; Barresi, G.; Barbaro, S.; Lombardo, G.; Rotolo, V.; Sebastianelli M.; Travagliato, G.; Palla, F. Fungi and bacteria in indoor cultural heritage environments: microbial-related risks for artworks and human health. Environ. Ecol. Res. 2016, 4(5), 257-264. https://doi.org/10.13189/eer.2016.040504
Thrasher, J.D. Fungi, bacteria, nano-particulates, mycotoxins and human health in water-damaged indoor environments. J. Comm. Pub. Health Nurs, 2016, 2(115), 2. https://doi.org/10.4172/2471-9846.1000115]. Fungal spores and metabolites (mycotoxins) that cause the development of upper and lower respiratory tract infections, allergic reactions. poisoning can cause bronchial irritation and allergy, broncho pulmonary mycoses, and hypersensitivity pneumonitis [Khan, A.H., Karuppayil, S.M. Fungal pollution of indoor environments and its management. Saudi J. Biol. Sci. 2012, 19(4), 405-426. https://doi.org/10.1016/j.sjbs.2012.06.002]. In chronic rhinosinusitis, fungal and bacterial biofilms are usually found, which can cause a chronic and antimicrobial-resistant stage of the disease [Shin, S.-H.; Ye, M.-K.; Lee, D.-W.; Geum, S.-Y. Immunopathologic role of fungi in chronic rhinosinusitis. Int. J. Mol. Sci. 2023, 24, 2366. https://doi.org/10.3390/ijms24032366]; the presence of mycotoxins in food and feed is dangerous due to their toxicity and carcinogenesis development [54,56];
- bacteria, as the main microorganisms present in the human body, are studied more widely, whereas databases of complete genomes of fungi present in the human body are less well provided [Lapiere, A., Richard, M. L. (2022). Bacterial-fungal metabolic interactions within the microbiota and their potential relevance in human health and disease: a short review. Gut Microbes, 2022, 14(1), 2105610. https://doi.org/10.1080/19490976.2022.2105610]; this lack of information underlies the development of resistant dysbiosis
- in the process of observed climatic changes, viral and bacterial diseases as a potential cause of epidemics and pandemics may fade into the background and give way to fungi, since it is fungi that can pose an equal or even greater threat: there are no vaccines against fungal pathogens yet, the arsenal of antifungal drugs is extremely limited, and fungi can live saprotrophic, producing a large number infectious spores, without requiring direct contact with the affected object, have a unique ability to adapt to new, including temperature conditions [Nnadi, N. E., & Carter, D. A. (2021). Climate change and the emergence of fungal pathogens. PLoS pathogens, 17(4), e1009503. https://doi.org/10.1371/journal.ppat.1009503]. Taking all this into account, fungal pathogens have recently been included for the first time in the "World Health Organization (WHO) fungal priority pathogens list", compiled from 19 groups of fungal microorganisms associated with a serious risk of human mortality or morbidity [Fisher M.C.; Denning D.W. The WHO fungal priority pathogens list as a game-changer. Nat. Rev. Microbiol. 2023, 21, 211–212. https://doi.org/10.1038/s41579-023-00861-x].
Thus, when studying the possibilities of preserving the sustainable development of mankind, fungi deserve no less attention than bacteria, especially as part of mixed consortia.”
- The paper covers a broad range of topics, including antimicrobial peptides (AMPs). Please specify and emphasize which pathogenic organisms are currently within the control range of AMPs.
AMPs are used against the following fungi: Candida, Cryptococcus, Aspergillus [Buda De Cesare G, Cristy SA, Garsin DA, Lorenz MC. Antimicrobial Peptides: a New Frontier in Antifungal Therapy. mBio. 2020, 11(6), e02123-20. https://doi.org/doi:10.1128/mBio.02123-20; Efremenko, E.; Aslanli, A.; Stepanov, N.; Senko, O.; Maslova, O. Various biomimetics, including peptides as antifungals. Biomimetics, 8(7):513, 2023], including those which are in the frame of biofilms [Fontanot, A.; Ellinger, I.; Unger, W.W.J.; Hays, J.P. A Comprehensive review of recent research into the effects of antimicrobial peptides on biofilms—January 2020 to September 2023. Antibiotics 2024, 13, 343. https://doi.org/10.3390/antibiotics13040343].
These references were added to the article.
In additions to the information, from my side I can say that Candida is at the first place as target for investigation of AMPs’ actions [Rodríguez-Castaño, G. P., Rosenau, F., Ständker, L., & Firacative, C. (2023). Antimicrobial peptides: avant-garde antifungal agents to fight against medically important Сandida species. Pharmaceutics, 15(3), 789; Freitas, Camila G., and Maria Sueli Felipe. "Candida albicans and antifungal peptides.Infectious Diseases and Therapy 12.12 (2023): 2631-2648].
AMPs are studied against Aspergillus fumigatus [Kelty, Martin T., Aracely Miron-Ocampo, and Sarah R. Beattie. "A series of pyrimidine-based antifungals with anti-mold activity disrupt ER function in Aspergillus fumigatus." Microbiology spectrum (2024): e01045-24] and Mucor circenelloides [Garcia, A., Huh, E. Y., Lee, S. C. (2023). Serine/threonine phosphatase calcineurin orchestrates the intrinsic resistance to micafungin in the human-pathogenic fungus Mucor circinelloides. Antimicrobial Agents and Chemotherapy, 67(2), e00686-22].
Pathogens of agricultural crops and mycotoxin producers are also in the area of attention of researchers [Han, Zhuoyu, Quirico Migheli, and Qing Kong. "Fusion Expression of Peptides with AflR Binuclear Zinc Finger Motif and Their Enhanced Inhibition of Aspergillus flavus: A Study of Engineered Antimicrobial Peptides." Journal of Agricultural and Food Chemistry (2024), Fan, L., Wei, Y., Chen, Y., Jiang, S., Xu, F., Zhang, C., et al. (2023). Epinecidin-1, a marine antifungal peptide, inhibits Botrytis cinerea and delays gray mold in postharvest peaches. Food Chemistry, 403, 134419]. These pathogens are a kind of test cultures, and the found or synthesized APMs can be used against a wider range of mycelial fungi and yeasts.
Of cause, it is not possible to add all references to the same text, so we were focused on the possible applications against fungi in the frame of mixed consortia since such biosystems are the most difficult for suppressing.
With high respect,
corresponding author -
Prof. Elena Efremenko

Reviewer 2 Report
Comments and Suggestions for Authors
This is a very comprehensive review discussing the microbial consortia from many challenging topics. However, the review could be more attractive and readable after some minor revisions.
- The introduction section is composed of too many paragraphs, it would be more attractive for the audience if the layout is more concise.
- The reviewer encouraged the authors to use some illustrations and figures from the cited papers to explain some microbial topics. That would make this review more attractive to the audience.
Author Response
On behalf of the authors, I would like to express my great gratitude to the Reviewer for the kind attitude towards the authors, a good assessment of our work and valuable suggestions for improving the quality of the review.
Comments and Suggestions for Authors from Reviewer #2
This is a very comprehensive review discussing the microbial consortia from many challenging topics. However, the review could be more attractive and readable after some minor revisions.
- The introduction section is composed of too many paragraphs, it would be more attractive for the audience if the layout is more concise.
Response: Part of the text in the Introduction section has been shortened, according to the recommendation of Reviewer 2.
- The reviewer encouraged the authors to use some illustrations and figures from the cited papers to explain some microbial topics. That would make this review more attractive to the audience.
Response: Several figures made by the authors have been added to the text of the article, which illustrate the possible types of biofilms formed with the participation of mycelial fungi and yeast, as well as the structures of individual antimicrobial agents discussed in the text, and mycoviruses. As a result, instead of three figures, there are 10 in the article, the numbering of the originally presented figures was changed due to the additions made.
With high respect,
corresponding author -
Prof. Elena Efremenko
